# Tea Tree Oil: Properties and the Therapeutic Approach to Acne—A Review

**DOI:** 10.3390/antiox12061264

**Published:** 2023-06-12

**Authors:** Tânia Nascimento, Diana Gomes, Ricardo Simões, Maria da Graça Miguel

**Affiliations:** 1Escola Superior de Saúde, Universidade do Algarve (ESSUAlg), Campus de Gambelas, Edifício 2, 8005-139 Faro, Portugal; 2Algarve Biomedical Center Research Institute (ABC-RI), Universidade do Algarve, Campus de Gambelas, Edifício 2, 8005-139 Faro, Portugal; 3Faculdade de Ciências e Tecnologia, Universidade do Algarve, Campus de Gambelas, 8005-139 Faro, Portugal; dianafofa2010@hotmail.com (D.G.); ricardo_simoes@hotmail.com (R.S.); 4Mediterranean Institute for Agriculture, Environment and Development, Mediterranean Institute for Agriculture, Environment and Development, Universidade do Algarve, Campus de Gambelas, 8005-139 Faro, Portugal

**Keywords:** acne vulgaris, antioxidant properties, antibacterial properties, anti-inflammatory properties, tea tree oil

## Abstract

Acne vulgaris is an inflammatory dermatological pathology that affects mostly young people. However, it can also appear in adulthood, mainly in women. It has a high psychosocial impact, not only at the time of active lesions but also due to the consequences of lesions such as scarring and hyperpigmentation. Several factors are involved in the physiopathology of acne and the constant search for active ingredients is a reality, namely phytotherapeutic ingredients. Tea tree oil is an essential oil extracted from *Melaleuca alternifolia* (Maiden & Betch) Cheel with known antibacterial, anti-inflammatory, and antioxidant properties, making it a candidate for the treatment of acne. This review aims to describe the various properties of tea tree oil that make it a possible ingredient to use in the treatment of acne and to present several human studies that have evaluated the efficacy and safety of using tea tree oil in the treatment of acne. It can be concluded that tea tree oil has good antibacterial, anti-inflammatory, and antioxidant properties that result in a decrease in the number of inflammatory lesions, mainly papules, and pustules. However, given the diversity of study designs, it is not possible to draw concrete conclusions on the efficacy and safety of this oil in the treatment of acne.

## 1. Introduction

Acne vulgaris is a skin disease that affects 80% of the teenage population; however, it can reach 100% because it is believed that there is no one who has not had a more or less severe episode of acne throughout their life. Although acne is not a life-threatening disease, it can have serious psychosocial consequences leading to low self-esteem, social isolation, and depression, as the severe disease can leave disfiguring scars on the face and is not limited to a few papules or comedones. *Acne vulgaris* is a chronic inflammatory disease in which the following actors are involved: increased sebum production and modification of its composition, increased proliferation and retention of hyperkeratosis, hyper-colonization of *Cutibacterium acnes* (formerly *Propionibacterium acnes*), *Staphylococcus aureus*, *Staphylococcus epidermidis*, and inflammation (Figure 1) [1,2].

The treatment of acne vulgaris is based on the type and severity of acne and takes into account the patient’s concomitant diseases and preferences, as specified in the European recommendation guidelines. The European guideline for the treatment of acne presents a very simple clinical classification of the pathology: 1. comedonal acne; 2. mild-moderate papulopustular acne; 3. severe papulopustular acne, and moderate nodular acne; 4. severe nodular acne and conglobate acne. More inflammatory lesions (papules and pustules) are found in higher severities of acne [3,4]. (Figure 2). Topical agents are recommended for mild to moderate comedogenic acne, especially retinoids (e.g., isotretinoin, adapalene), benzoyl peroxide, and azelaic acid. Retinoids act as suppressors of comedogenesis, sebum production, and inflammation, and as normalizers of epithelial desquamation. Benzoyl peroxide has antibacterial and anti-inflammatory effects and shows mild comedolytic activity. Azelaic acid also has antimicrobial, anti-inflammatory, and comedolytic properties [4,5].

For mild to moderate papulopustular acne, the use of fixed combinations of benzoyl peroxide with adapalene or benzoyl peroxide with clindamycin is fully recommended. In more severe cases, topical retinoids, especially adapalene, can be combined with systemic antibiotics (the macrolides erythromycin, clindamycin, azithromycin, and the tetracyclines doxycycline, minocycline, and lymecycline). In women, administration of antiandrogenic hormone therapy in conjunction with systemic antibiotics and/or topical treatments other than antibiotics may also be considered [4,5]. Adverse effects, inadequate response to therapy, the high cost of some acne treatments, and the possible emergence of bacterial resistance to antibiotics are leading to an increasing demand for alternative and complementary therapies, especially those of natural origin [5,6,7]. Moreover, in natural origin, it has been investigated whether antibiotic substitutes with anti-inflammatory activity inhibit the growth of *C. acnes* without the disadvantage of skin irritation, dry skin, or other side effects, as is the case with some antibiotics used to treat acne. Lactic acid bacteria such as *Lactobacillus paraplantarum* THG-G10, isolated from traditional kimchi in the Republic of Korea, have an antibacterial effect against *C. acnes* [8].

Herbal therapy for acne has been promoted because of its long history of use, fewer side effects, better tolerability for patients, and relative cost-effectiveness. Many medicinal plants that have long been used in traditional cultures have entered the growing market of “cosmeceuticals” [9,10]. Phenolic compounds, terpenoids and steroids, alkaloids, and fatty acids are only some examples of a great group of phytoconstituents with antimicrobial, antioxidant, anti-inflammatory, and anti-androgen effects [9]. Essential oils (EOs), lipophilic and volatile secondary plant compounds with molecular weights below 300, are extracted from buds, flowers, roots, barks, fruits, seeds, twigs, wood, and stems by steam distillation, by mechanical processes from the epicarp of citrus fruits or by dry distillation [11] These EOs are believed to have antioxidant, antimicrobial, antiprotozoal, antiviral, antifungal, and antineoplastic properties [12].

Considering the diverse properties of TTO, its use in the therapeutic approach to acne is relatively common. Thus, the aim of this review is to describe the composition of TTO and its properties, as well as to describe, characterize and discuss trials conducted in humans using TTO in the treatment of acne, demonstrating the efficacy and safety of TTO in this pathology.

## 2. Physiopathology of Acne Vulgaris

As already mentioned, there are several factors related to the development of acne, namely hyperkeratinization, hyperseborrhea, and bacterial colonization. As a result of the interaction between these factors, an inflammatory process may occur [12]. Initially, and clinically invisible, a microcomedon develops in the pilosebaceous unit that can evolve into clinically visible lesions: non-inflammatory lesions (open comedones—blackheads and closed comedones—whiteheads) that can later develop inflammatory lesions such as papules and pustules [13,14]. Comedon formation results from the abnormal proliferation and differentiation of undifferentiated sebocytes. These, instead of differentiating into mature sebocytes, differentiate into sebaceous duct cells and infundibular keratinocytes, leading to abnormal keratinization and their retention in the follicular channels. Several factors are involved in this process, such as changes in the production and composition of the lipids that make up sebum, androgens, cytokine production, retinoid control, and the presence of bacteria present in the microbiome [13,14,15,16]. Even microcomedons (invisible), seem to show bacterial colonization [17].

Microbiome refers to microorganisms (bacteria, viruses, and fungi) and their environment, and has an essential role in the maintenance of health and the occurrence or progression of disease. The skin microbiome is divided into commensal or resident skin microbes and transient microbes. The resident microbiome includes *C. acnes* and *Staphylococcus epidermidis*, while the transient microbiome includes *Staphylococcus aureus* [18,19]. Bacterial composition depends on several factors. Intrinsic factors such as gender, age, skin type or endocrine factors, and external factors such as environmental factors, use of cosmetics or make-up, smoking, diet, and ultraviolet radiation exposure, among others [20,21]. Microbial imbalance or ‘dysbiosis’, compared to the normal distribution in healthy tissues, is thought to be involved in the pathophysiology of inflammatory acne [18,19,20]. To maintain skin health and homeostasis, *C. acnes* and *S. epidermidis* produce short-chain fatty acids and peptides, respectively, nevertheless, these microbial species also appear in opportunistic infections [19]. The virulence of *C. acnes* seems to be influenced by environmental and endocrine factors. However, other microorganisms also seem to be involved in acne, such as *C. granulosum*, more virulent than *C. acnes*, or *Malassezia* which seems to contribute to hyperkeratinization and comedon formation [22].

Although the etiologic step process of acne has considered inflammation as a response to the overgrowth of *C. acnes*, *S. aureus*, and *S. epidermidis*, it has recently been hypothesized that inflammation precedes the formation of *C. acnes* and biofilm, and before the formation and maturation of comedones [19,23,24].

It has been accepted that abnormal desquamation of the follicular epithelium can lead to blockage of the follicular canal. The sebum retained inside leads to the proliferation of *C. acnes*. The lipase of this microorganism hydrolyzes the triglycerides of sebum into free fatty acids, which are both comedogenic and proinflammatory. In addition, *C. acnes* also releases various chemotactic products (Figure 3) that attract neutrophils. The lysosomal enzymes released by the neutrophils in turn cause a tear in the follicular wall with subsequent leakage of lipids and keratin into the surrounding dermis, triggering intense acute and chronic inflammation [25].

Recently, several studies on *C. acnes* have led to the classification of subspecies or phylotypes (IA1, IA2, IB, IC, II, and III) based on the comparison of their genomic sequences. Each of them has a unique set of characteristics, activities, and pathogenic potential. A skin metagenomic analysis comparing patients with acne to healthy individuals revealed specific genomic elements in disease-associated *C. acnes* strains [13,26]. Analysis of host immune responses against different *C. acnes* strains also confirmed strain-level differences. Thus, health-associated *C. acnes* strains induced higher levels of the anti-inflammatory IL-10, whereas acne-associated *C. acnes* strains induced higher levels of the proinflammatory cytokines interferon (IFN)-γ and interleukin (IL)-17 in peripheral blood mononuclear cells, which are present in greater amounts in acne lesions. The development of acne appears to be associated with a loss of *C. acnes* phylotype diversity, with a greater predominance of the *C. acnes* phylotype IA1. *C. acnes* also increased the expression of filaggrin and integrin (Figure 3), which affect abnormal adhesion and differentiation of keratinocytes [13,26]. A recent study demonstrated the induction of pro-inflammatory cytokine expression by *C. acnes*, nevertheless, the inflammatory response was increased in the presence of particles found in tobacco smoke or dust, called particulate matter [27].

*Cutibacterium acnes* can grow in macrocolonies, particularly phylotypes IA subtypes A1 and A2, through the production of a biofilm in the hair follicles, which is greater in acne patients than in healthy ones. Such capacity leads to increasing keratinocyte cohesion. Moreover, in acneic skins, it was found that *C. acnes* strains express virulence genes not detected in those strains growing in healthy skins [13]. Therefore, phylotype IA1 can produce virulence factor proteins that trigger to host-tissue degradation and inflammation as well as host-interacting factors, such as Christie Atkins Munch Petersen (CAMP) factors, hemolysins, and dermatan sulphate-binding adhesins (DsA1 and DsA2) [28]. This virulence gives rise to a microenvironment of the bacterium with an excess of sebum and its change in sebum content composition, along with biofilm formation [13]. *Cutibacterium acnes* biofilms and virulence factors have been detected in pilosebaceous follicles in biopsies taken from patients with acne. In these patients, there is a secretion of propionic acid by *C. acnes*, leading to the formation of keratinocytes with irregular cellular morphologies. This differentiation gives rise to the development of inflammatory acne lesions and the formation of microcomedones [12]. So far, there is not in vivo evidence of Quorum Sensing (QS) by *C. acnes* but in vitro studies have demonstrated increased production of autoinducers (AI-2) by *C. acnes* derived from human acne isolates and their levels were three times higher in mature *P. acnes* biofilms than in planktonic cells, with an increase of lipase activity in biofilms [29]. 

It is observed that the microbiome, especially the presence of *C. acnes*, and inflammation, generated by *C. acnes* but also by external factors, play a very important role in the development of inflammatory lesions and can be targeted in the therapeutic approach to the pathology.

## 3. Components of Tea Tree Oil (TTO) and Their Variability

*Melaleuca alternifolia* (Maiden & Betch) Cheel is an evergreen Australian native tree and belongs to the Myrtaceae family. The leaves and branches of this species are rich in EOs, which along with other *Melaleuca* species (*M. dissitiflora* Smith, *M. linariifolia* F. Mueller, and *M. uncinata* R. Br.), EOs are known as tea tree oil (TTO). This TTO is used in the pharmaceutical, cosmetic, and food industries due to its antimicrobial, antioxidant, anti-inflammatory, and antineoplastic properties [30,31,32]. Adulteration of TTO happens, which may be related to confusion with the common name (tea tree) of some species in the genus *Leptospermum*, and species in the genera *Kunzea* and *Baeckea* from Australia and New Zealand, which also have the same common name. Nevertheless, the financial gain can be the most important factor in the adulteration of TTO [31], through the addition of lower-cost byproducts derived from *Eucalyptus* essential oil or even through the addition of pure chemical compounds obtained by chemical synthesis or fermentation [33].

The Australian tea tree industry exports 90% of its production to international markets, around 900 metric tonnes of oil per annum. The production is mainly based on genetically improved populations resulting from a long-term breeding program [34].

There are six oil chemotypes in *M. alternifolia* identified by the presence of distinct percentages of terpinen-4-ol, 1,8-cineole, and terpinelone: terpinen-4-ol chemotype containing 30 to 40% of this monoterpenoid (used in commercial TTO production); terpinolene chemotype; 1,8-cineole chemotype; and three chemotypes all dominated by 1,8-cineole but differing in either terpinen-4-ol or terpinolene content [30,35] According to the International Standard Organization (ISO 4730:2017) [36], TTO must have a minimum terpinen-4-ol content of 35% and a maximum 1,8-cineole content of 10%, nevertheless market prefers a TTO with the highest terpinen-4-ol (higher than 38%) and the lowest 1,8-cineole content (less than 3%) possible [36,37,38]. Ten monoterpene/monoterpenoids and five sesquiterpene/sesquiterpenoids can be found in TTO (Figure 4).

To avoid adulterated TTO with other compounds, the revised international standard, ISO 4730-2017 [36] included a chiral ratio parameter, indicating that oil from the terpinen-4-ol chemotype must have a chiral ratio of 67–71% for (+)-terpinen-4-ol and 29–33% for (−)-terpinen-4-ol. Southwell et al. [39] characterized the enantiomeric ratios of the key chiral monoterpenes limonene, terpinen-4-ol, and α-terpineol in the 6 most common chemotypes of *M. alternifolia*. The study found that there was a difference among the average chiral ratios of the 6 chemotypes of this species. So, the authors concluded that oil blended from more than one chemotype has the potential to change the chiral ratio from a chemotype average.

In the same chemotype of TTO, there is chemical variability, as can be observed in Figure 4, since it is permitted an interval of percentages for each volatile compound. This variability can be attributed to genetic [40,41] and environmental factors [35,40], leaf age and mechanical damage [42], and harvesting season [43], hydrodistillation-induced artifact formation [44]. Other isolation procedures [45,46] have been used for isolating the volatiles of tea trees, which may contribute to the chemical variability, but they are not considered in the present review, due to the definition of ISO [11] for EOs. 

## 4. Biological Properties of TTO

According to the review made by Carson and Riley [47], TTO was applied for many purposes: perionychia, empyema, gynaecological conditions, epidermophyton infections, impetigo contagiosum, pediculosis, ringworm, tinea, throat, psoriasis, and mouth conditions. A systematic review to assess preclinical and clinical studies focused on the antiparasitic activity of TTO against *Demodex* mites, scabies mites, house dust mites, lice, fleas, chiggers, and bed bugs revealed the efficacy of TTO and its components against ectoparasites of medical importance. Such results can justify the use of TTO in the pharmacotherapy of ectoparasitic infections [48] with the advantages to be simple to use and having better side effects than the current treatments [49].

In Australian traditional medicine, Aboriginal people used *M. alternifolia* to treat bruises, insect bites, and skin infections [50]. During World War II, TTO was used as an insect repellant to reduce the infection rate following skin injuries [47,50]. However, after that period, a declining use was observed owing to unreliable supply and variable quality [47]. Currently, to prevent this variability, the ISO has established limits for the proportions of volatile constituents that must be present in any product(s) marketed as the terpinen-4-ol chemotype of TTO. This norm also includes the enantiomeric ratio of terpinen-4-ol [36,51].

The ethnopharmacological studies made on the utilization of TTO may disclose their antimicrobial, antioxidant, and anti-inflammatory activities. For this reason, a brief review of these properties made in in vitro assays is compiled in the present work.

### 4.1. Antimicrobial Activity of TTO

*Melaleuca alternifolia* essential oil has antibacterial, antifungal, and antiviral activities, and such properties have been attributed to terpinen-4-ol. For example, May et al. [52] reported that a terpinen-4-ol-rich TTO was more effective against several multi-drug-resistant organisms, including MRSA, glycopeptide-resistant enterococci, aminoglycoside-resistant *Klebsiella*, *Pseudomonas aeruginosa* and *Stenotrophomonas maltophilia*, and also against sensitive microorganisms than a TTO with a lower percentage of that monoterpenoid and higher percentage of 1,8-cineole. However, the antimicrobial activity of TTO that can be attributed to the loss of membrane integrity and function, letting the intracellular material out, disabling to maintain homeostasis, and inhibiting respiration [30,53] is not only attributed to the amount of terpinen-4-ol, but from to the complex interaction among different components such as 1,8-cineole, α-terpinene, γ-terpinene, terpinolene, among other ones of the TTO [28,38]. The antimicrobial activity of ten EOs of terpinen-4-ol chemotype assessed against *Candida glabrata*, *Herpes simplex* virus type 1 (HSV-1), methicillin-resistant *Staphylococcus aureus* (MRSA), and *Pseudomonas aeruginosa* grown in planktonic mode or biofilms showed that only five had significant antimicrobial activity by reducing bacterial survival in biofilms, generating oxidative damage in *C. glabrata*, and decreasing HSV-1 infectivity [28]. The authors were not able to correlate such activities to the amounts of terpinen-4-ol in the sample oils.

In *Escherichia coli*, *S. aureus*, and *C. albicans*, Cox et al. [54] concluded that the antimicrobial activity of TTO results from its ability to disrupt the permeability barrier of microbial membrane structures, that is, similar to that of other disinfectants and preservatives, such as phenol derivatives, and chlorhexidine. Nevertheless, this observation was more evident when *E. coli* cells were in the exponential phase because, at the stationary phase, *E. coli* showed an increased TTO tolerance [55,56]. According to the authors, such results can be partially explained by alterations in membrane/structure that occur during the stationary phase. The disruption of the cell walls and membranes were also reported by Li et al. [57]. They observed that TTO penetrated through the cell walls and cytoplasmic membranes of *E. coli*, *S. aureus*, *C. albicans,* and *Aspergillus niger*, damaging these structures with subsequent loss of cytoplasm content and cell death. Cuaron et al. [58] reported that TTO was able to denature proteins, alter the membrane, cell wall structure, and function of *S. aureus*. In addition, TTO was able to down-regulate the genes involved with energy-intensive transcription and translation and alter the regulation of genes involved with heat shock and cell wall metabolism in *S. aureus*. For example, the inactivation of those heat shock genes, which encodes a regulatory system that responds to peptidoglycan biosynthesis inhibition in *S. aureus*, led to an increase in TTO susceptibility [58].

Although the absence of a correlation between the terpinen-4-ol amount and the biological activity of TTO, the majority of the antimicrobial activities are made with the terpinen-4-ol chemotype against several microorganisms: Table 1 depicts the antimicrobial activity of TTO found in diverse works and against several microorganisms.

*Legionella pneumophila* is responsible for severe pneumonia named Legionnaires’ disease [58]. Mondello et al. [59] tested TTO and terpinen-4-ol against diverse strains of *L. pneumophila* and reported that both had activity against this microorganism (Table 1). In addition, the authors also found that the activity was temperature-dependent, that is, the values of minimal inhibitory concentration (MIC) and minimal bactericidal concentration (MBC) decreased with increasing temperature (36 to 45 °C).

One of the main causes of endodontic failure is persistent root canal infection and *Enterococcus faecalis,* a Gram-positive facultative anaerobe, is associated with this persistent endodontic infection. Chlorhexidine (CHX) has been used to eliminate *E. faecalis* from superficial layers and dentinal tubules if applied for seven days. Even so, the microorganism is able to survive harsh environments and develop resistance to CHX and antibiotics as well [60]. Qi et al. [60] evaluate the antimicrobial effects of TTO on planktonic *E. faecalis* and biofilms compared with 0.2% CHX. The results are compiled in Table 1. In what concerns the biofilm formation, the effects of TTO on pre-formed *E. faecalis* biofilm, at 12, 24, and 48 h time points there was no difference between 2%, 1%, 0.5% TTO, and the CHX group for12 h, 24 h or 48 h. However, when the TTO concentration was ≤0.25%, the activity was statistically different from the CHX group at all time points [60]. The results obtained by these authors permitted them to conclude that TTO could inhibit *E. faecalis* by destroying cell membranes, inhibiting the formation of *E. faecalis* biofilms, and eliminating mature-formed biofilms.

The biofilm is a structured community of microorganisms, enclosed in a self-produced polymeric matrix, able to adhere to an inert or living surface in an aqueous medium. In hospital-acquired infections, many times the bacteria involved in the infection are found within biofilms. These may have a single microbial species or a set of microorganisms (including fungi, algae, and protozoa) [61]. The authors [61] studied the effect of two EOs being one of them TTO, against both mature biofilms and biofilms in the process of formation, produced by vancomycin-resistant enterococci (VRE), methicillin-resistant *Staphylococcus aureus* (MRSA) and broad-spectrum-lactamase-producing *Escherichia coli* (ESBL). The MIC values are presented in Table 1. However, the authors also determined the MIC of the associations of TTO and other EO, and the combinations of TTO and antibiotic (not depicted in Table 1). Their results confirmed a synergistic effect among the EOs and antibiotics. The same synergistic effect among the EOs and antibiotics was also observed for the biofilm formation and the anti-mature biofilm activity [61].

Not only liquid EOs but also volatile forms of EOs show antimicrobial activity. The utilization of the volatile form of EOs has the advantage to provide higher concentrations of active compounds to the infection site and avoid the side effects of the systemic administration of EOs [62]. Taking this in mind, the authors [62] evaluated the antimicrobial and antibiofilm in vitro activity of seven volatile and liquid EOs against MRSA and methicillin-sensitive (MSSA) clinical and reference strains, generally responsible for wound and bone infection. Table 1 only depicts the results of TTO. Liquid and volatile thyme EO had the highest antibiofilm activity in contrast to the vapor phases of tea tree and lavender. The antimicrobial activity of the vapor phase of TTO was also performed and compared with the liquid phase against methicillin-sensitive *Staphylococcus aureus* (MSSA), *Escherichia coli*, and clinical strains of methicillin-resistant *S. aureus* (MRSA), extended-spectrum beta-lactamases producer carbapenem-sensitive *Klebsiella pneumoniae* (ESBL-CS-Kp), carbapenem-resistant *K. pneumoniae* (CR-Kp), *Acinetobacter baumannii* (CR-Ab), and *Pseudomonas aeruginosa* (CR-Pa) [63]. Moreover, synergistic activity between TTO and different antimicrobials were also determined. TTO showed bactericidal activity against all the tested microorganisms (Table 1). TTO in combination with oxacillin showed a high level of synergism at sub-inhibitory concentrations, against MRSA. The vapor phase assay showed high activity of TTO against CR-Ab. According to these results, the authors [63] suggest that TTO in the vapor phase might represent a promising option for local therapy of pneumonia caused by CR-Ab.

*Pseudomonas fluorescens* can form biofilm in soil and water habitats, can influence food spoilage, water quality, and plant diseases, and create nosocomial infections [64]. *Salmonella enterica* is the main cause of acute foodborne illness and is a source of infection from poultry meat, and can form biofilm even in low temperatures. Moreover, the biofilm can be formed on glass and wooden surfaces, which can cause cross-contamination of vegetables [64]. Antibiofilm activity was observed against *P. fluorescens* and *enterica* through the observation of degradation of the protein spectra using the matrix-assisted laser desorption/ionization—time-of-flight mass spectrometer (MALDI-TOF MS) and under the effect of TTO [64]. TTO was also used on Gram-positive, and Gram-negative bacteria, and yeasts, and the values found for the growth inhibitions are depicted in Table 1.

Banes-Marshall et al. [65] investigated the effect of TTO on diverse isolates from leg ulcers, pressure sores, skin, and vagina. The MIC and MBC values are resumed in Table 1. According to the results, *S. aureus* and *Candida* spp. were particularly sensitive to the action of TTO, and therefore, it may have a positive role in the growth inhibition of the commonly isolated wound pathogens, and in the frequent infection in immunosuppressed and antibiotic-treated patients [65].

The activity of TTO against twenty-seven clinical isolates of *S. aureus* and the reference strain *S. aureus* NCTC 8325-4 were assayed [64] either in planktonic cells or biofilm (Table 1). The killing rate for stationary phase cells was less affected by increasing TTO concentration than that for exponential phase cells. Moreover, the fastest killing of biofilm occurred during the first 15 min of contact with TTO, and concentrations above 1% did not affect the results [66].

Loughlin et al. [67] compared the bactericidal activity of the racemic terpinen-4-ol and the L-isomer terpinen-4-ol with TTO manufactured by two enterprises, against clinical skin isolates of MRSA and coagulase-negative staphylococci (CoNS). The MIC values are depicted in Table 1. Terpinen-4-ol was a more potent antimicrobial against MRSA and CoNS isolates than the TTO. In any case, the concentrations tested displayed toxicity to human fibroblast cells [67].

The antimicrobial effect of TTO against *S. aureus*, *E. coli*, and *C. albicans* was also studied by Blejan et al. [68] (Table 1). These authors compared the activity of TTO with other EOs and concluded that oregano and basil EOs had a similar antimicrobial effect against *S. aureus*, while against *E. coli*, the antimicrobial activity was similar for oregano, basil EO, and TTO. Against *C. albicans*, TTO and basil EO showed strong antifungal properties [68]. According to these authors, linalool might be responsible for the activity against *S. aureus*, whereas, for *E. coli*, the effect could be attributed to linalool, 1,8-cineole, terpinen-4-ol, α-pinene, *p*-cymene, α/γ-terpinene, whereas for *C. albicans*, 1,8-cineole, and terpinen-4-ol could be responsible for the activities found.

Silver nanoparticles (AgNPs) have displayed antimicrobial effects against a wide range of microorganisms, including antibiotic-resistant strains. Several methods have been used to synthesize AgNPs. The green synthesis method, in which aqueous plant extracts are used, provides a simple, cheap, fast, energy-efficient, and eco-friendly alternative to the traditional chemical and physical methods of nanoparticle synthesis [69]. Ramadan et al. [69] studied the antimicrobial potential of TTO and AgNPs (obtained by using an aqueous extract of *M. alternifolia*) against selected skin-infecting pathogens, including bacteria, fungi, and viruses (Table 1). AgNPs had better antiviral activity against both herpes simplex virus type 1 (HSV-1) and herpes simplex virus type 2 (HSV-2) than TTO at its maximal noncytotoxic concentration (0.043%, *v/v*).

The activities of TTO against lactobacilli and a range of organisms associated with bacterial vaginosis were evaluated by Hammet et al. [70]. Table 1 lists MIC data. It is possible to observe that all lactobacilli tested were appreciably more resistant to TTO than organisms known to be associated with bacterial vaginosis, with at least a twofold difference in MIC values. Three batches of TTO were able to inhibit the growth of *C. acnes*, with a MIC value of 0.25% (*v/v*) (Table 1) [71]. 

Nenoff et al. [72] evaluated the antifungal activity of TTO against several dermatophyte and yeast strains, among the latter 31 strains of the *Malassezia furfur,* and determined the MIC values (Table 1). 

Ngeng et al. [73] evaluated the anti-quorum sensing activity of TTO through the violacein production in *Chromobacterium violaceum*. The quorum sensing inhibition diameter zone in *C. violaceum* was 14.3 ± 0.5 mm at MIC concentration (Table 1). However, *Citrus sinensis,* also studied by these authors, presented better activity than TTO, since it could inhibit violacein production right down MIC/16 whereas TTO did not inhibit violacein production beyond MIC/4 [73]. Violacein is an antioxidant that protects the bacterial membrane against oxidative stress. When TTO inhibits violacein production and swarming motility, it can suppress virulence and reduce biofilm risks during infections [73].

The synergistic effect between some EOs has also been performed through the Fractional Inhibitory Concentration Index (FICI) of the binary combinations of EOs determined by the checkerboard method. Following this method, and evaluating the MIC and Minimum Bactericidal Concentration (MBC), the authors concluded that TTO/lavender oil mixtures showed a synergistic effect against *Streptococcus pyogenes* and *Streptococcus agalactiae*; TTO/oregano oil had a synergistic effect against *Staphylococcus aureus* and *S. agalactiae*. According to these results, the authors [74] concluded that combination against pathogens should be preferred as potential antimicrobial agents. Combinations of commercial EOs mainly applied in aromatherapy for respiratory tract infections were also studied by [75] aiming the antimicrobial, anti-inflammatory, and toxicity properties. Five combinations were found presenting antimicrobial activity, reduced cytotoxicity, and improved anti-inflammatory effects, having four the presence of TTO: *Cupressus sempervirens* L. + *M. alternifolia*; *Origanum marjorana* L. + *M. alternifolia*; *Myrtus communis* L. + *M. alternifolia*; *Origanum vulgare* L. + *M. alternifolia* at 1:1 ratios. The microorganisms included the Gram-positive strains *Staphylococcus aureus* (ATCC 25924), *Streptococcus agalactiae* (ATCC 55618), *Streptococcus pneumoniae* (ATCC 49619), and *Streptococcus pyogenes* (ATCC 12344); the Gram-negative strains *Haemophilus influenzae* (ATCC 19418), *Klebsiella pneumoniae* (ATCC 13883) and *Moraxella catarrhalis* (ATCC 23246); and the non-pathogenic *Mycobacterium* strain M. *smegmatis* (ATCC 19420) and yeast strain *Cryptococcus neoformans* (ATCC 14116) [75].

Abdelhamed et al. [76] reported that TTO (Table 1), thyme and clove EOs have the capacity for inhibiting the growth of *C. acnes* and *Staphylococcus epidermidis*, although thyme essential oil was more effective since it was able to eliminate the initial bacterial inoculum after 10 h and 6 h of exposure for *C. acnes* and *S. epidermidis*, respectively. In contrast, Esmael et al. [77] reported that TTO was more effective than rosemary oil as a growth inhibitor of three groups of the acne-inducing bacteria *C. acnes* EG-AE1, *S. epidermidis* EG-AE2 and *S. aureus* EG-AE1, from Egypt. The chemical composition of TTO described by both teams was distinct, Abdelhamed et al. [76] described that 4-terpinenyl acetate dominated the EO extracted from the buds and terpinen-4-ol was absent. TTO used by Esmael et al. [77] for the determination of antimicrobial activity did not present those compounds.

Invasive fungal wound infections reported in trauma patients cause considerable morbidity and mortality despite the standard of care treatment in trauma centers [70]. Homeyer et al. [78] assessed the activity of various concentrations of TTO (without chemical composition provided) against 13 clinical filamentous fungal isolates comprising nine species (*Exophiala* sp., *Apophysomyces* sp., *Aspergillus fumigatus*, *Aspergillus flavus*, *Aspergillus terreus*, *Actinomucor* sp., *Mucor* sp., *Fusarium* sp., and *Absidia* sp.). Seven concentrations of TTO were assayed (100%, 75%, 50%, 25%, 10%, 5%, and 1%). For the majority of isolates, fungicidal activity was observed for all of the concentrations tested, 1, 10, and 100% (*v/v*), following 12–24 h exposures. *Aspergillus terreus* and *Absidia* spp. were more tolerant to the activity of TTO requiring higher concentrations for greater log-reductions. Cell viability assays were also performed in vitro using human fibroblasts, keratinocytes, osteoblasts, and umbilical vein endothelial cells. The activities found were dose-dependent with significant cytotoxicity at concentrations of ≥10% [78]. Despite these results, a recent meta-analysis showed that TTO had a limited antifungal effect (*Trichophyton rubrum*, *Trichophyton mentagrophytes* complex, and *C. albicans*) when compared to other EOs (*Cinnamomum zeylanicum* and *Thymus vulgaris*) [79]. Thus, the use of TTO against dermatophytes would not be advisable.

As can be seen in Table 1 there was a wide range of MIC values for the antimicrobial potency, which can be due to the diverse bacterial species and strains used, their culture methods, and the chemical composition of the TTO samples, and previously reported [80] in a compilation about the utilization of fifteen plant-based natural compounds on the antimicrobial activities.

#### Tea Tree Oil Formulations with Antimicrobial Activity

Carson et al. [81] in a revision, and May et al. [52] reported that microorganisms colonizing skin transiently were more susceptible to TTO than commensal microorganisms. Such results would be promising since the commensal microorganisms would constitute a barrier against the development of pathogenic microorganisms. The in vitro antimicrobial activities of TTO have led to the development of formulations to achieve better biological activities, as compiled below in a brief way.

The antimicrobial activity of TTO is widely described, but allergic contact dermatitis in susceptible individuals can occur, particularly if the oil is old, which can happen during the daily opening of bottles where TTO is kept [82]. So, an alternative to overcome this inconvenience could be of interest. Minghetti et al. [82] developed a patch (monolayer device) prepared by using methacrylic copolymers, Eudragit E100 (EuE100) or Eudragit NE (EuNE), and a silicone resin, BioPSA7-4602 (Bio-PSA). TTO and oleic acid (skin penetration enhancer in patches) contents were fixed at 10% *w/w* and 3% *w/w*, respectively. The patches were prepared by a casting method and characterized in terms of terpinen-4-ol amounts and skin permeability. The patches were self-adhesive controlled release matrix with TTO and a removable protecting foil [82]. The skin permeation of terpinen-4-ol enhanced when TTO was used in combination (1:1) with oleic acid, being up to 10-fold higher than pure TTO. This ability of oleic acid of enhancing the penetration of TTO can be a kick-off for the optimization of the efficacy and safety of TTO patches [82].

A TTO lipid-based nano-formulation (TTO-LNF) was developed [83] using a quality-by-design (QbD) approach. Using a mixture experimental design, TTO-LNF was optimized with 5% TTO, 10% surfactant (Kolliphor. RH40:Tween-80 = 50:50, *w/w*), 5% co-surfactant (Transcutol P), and 80% water. To make easier the topical administration, it was formulated a TTO-LNF gel adding xanthan gum. The in vitro antibacterial tests of TTO-LNF, TTO-LNF, TTO-LNF, and 5% TTO solution against *Staphylococcus epidermidis* ATCC 35984 and *Pseudomonas aeruginosa* PAO1 were performed. The results showed that *P. aeruginosa* was more susceptible than *S. epidermidis* to the TTO-LNF gel. The respective inhibition zones were 7.8 mm and 4.2 mm. Similarly, the TTO-LNF showed 6.4 mm for P. aeruginosa and 5 mm for *S. epidermidis*. The formulation was also shown to be more effective than the 5% TTO solution. The bacterial growth curve conducted over time showed that the treatment of *S. epidermis* with TTO-LNF gel and TTO-LNF had a notable suppression of bacteria growth for 24 h, even better than the antibiotic kanamycin. The antibacterial effects of blank LNF and blank LNF gel (without TTO) can be explained by the antimicrobial activity of Tween 80 (5%) [83].

Hydrogels are biodegradable three-dimensional crosslinked polymer networks able to assimilate large amounts of water or biological fluids and provide controlled release of drugs [84]. The remarkable water absorption capacity of these systems is due to the high content of functional hydrophilic groups (carboxyl, hydroxyl, and amino groups) contained in the polymer [85]. The polymers can be synthetic or natural (collagen, gelatin, polydopamine, elastin, chitosan, hyaluronic acid, alginate, and cellulose) [85]. Low et al. [84] used chitosan to fabricate hydrogels combined with TTO and silver ions (Ag^+^) to treat common wound-infecting pathogens. Silver ions are recognized as possessing antibacterial, antiviral, antiprotozoal, and antifungal activity [84]. The hydrogels loaded with TTO and Ag^+^ displayed antimicrobial activity against *P. aeruginosa*, *S. aureus*, and *C. albicans*. The combination lowered the effective concentrations needed for the antimicrobial activity. According to the results obtained, the authors [84] proposed that the relationship between the variables in the fabrication of hydrogels requires deeper studies to be used as antimicrobials in the treatment of acute wounds. This approach will achieve smarter delivery systems [84].

Ghosh et al. [86] fabricated a hydrogel scaffold with β-cyclodextrin and chitosan along with pectin, carboxymethyl cellulose, and polyethylene glycol 400 (PEG 400) as a plasticizer, and copper sulfate as crosslinker loaded with TTO, to enhance the antimicrobial activity in the treatment of acne and other skin infections. These scaffolds could be used as sheet masks or patches. Two hydrogels were formulated differing in the ratio of pectin, carboxymethyl cellulose, chitosan, and β-cyclodextrin (1:1:0.8:0.8 or 1:1:0.8:0.1) plus the plasticizer and the crosslinker. At the moment of the antimicrobial assay, 10 μL of TTO were infused in the scaffold which was in the hole of a plate. *Pseudomonas* sp. was the microorganism used in the assay. Only the first formulation with TTO presented better physical characteristics along with the best antibacterial activity [86]. It could also deliver TTO in a more efficient way enhancing, therefore, the antibacterial activity. The authors [86] concluded that this hydrogel can be used as a sheet mask to deliver TTO, for treating infection and acne.

Antimicrobial in situ-forming alginate wound dressing with TTO microemulsions was assayed in which alginate hydrogels were prepared by a layer-by-layer spray deposition. This formulation would be useful as an advanced dressing for infected wounds [87]. Diverse combinations of TTO, water, polysorbate 80, and ethanol were tested and through the pseudoternary phase diagrams, it was possible to find a stable spherical microemulsion with TTO at 20%, with good antimicrobial activity. The antimicrobial effect of alginate/TTO microemulsion hydrogels on *Escherichia coli* strains was remarkable. Catanzano et al. [87] considered that such a formulation has the potential to as a bioactive wound dressing.

Emulgels with jelly-like consistency are used in dermatological products due to their better applicability, thixotropic behavior, greaseless nature, improved spreadability, and controlled rheological properties. They present the properties of both emulsions and gels [88]. Sinha et al. [88] optimize a nanoemulsion-based emulgel formulation as a vehicle for topical delivery of TTO. The central composite design was used to choose the best processing conditions for nanoemulsion preparation by high energy emulsification method, namely surfactant concentration (Tween^®^ 20), co-surfactant concentration (Cremophor EL^®^), and stirring speed. The TTO concentration used was 5% (*v/v*) dissolved in Cremophor EL^®^. After optimization, the nanoemulsion was converted into emulgel using the polymer Carbopol 940 and triethanolamine as an alkalizer. The antimicrobial activity of the emulgel was assayed against the following microorganisms: *Staphylococcus aureus* MTCC-96, *Streptococcus mutans* MTCC-890, *Pseudomonas aeruginosa* MTCC-741, *Escherichia coli* MTCC-723, *Candida albicans* (wild) MTCC-1637, and *Candida albicans* (CA) AIIMS. The results were compared with those assays using a conventional gel and pure TTO. The emulgel revealed broader zones of growth inhibitions than conventional gel or pure TTO [88].

Other TTO delivery system was studied [89] for combating Gram-positive and Gram-negative bacteria and to be used as an antimicrobial and healing agent in skin wounds. Semisolid bicontinuous microemulsions containing TTO is an example, in which a formulation consisting of Kolliphor^®^ HS 15 (31.05%), Span^®^ 80 (3.45%), isopropyl myristate (34.5%), and distilled water (31%) with TTO incorporated in the proportion of 3.45% (*v/v*) was selected after optimization through diagram construction [89]. In vivo studies using male and female Swiss mice (*Mus musculus* Linnaeus, 1758) showed that the TTO-loaded bicontinuous microemulsion was effective in the healing process of skin wounds because it promotes a higher percentage of wound edge contraction. Antibacterial activity for Gram-positive and Gram-negative bacteria was also observed. According to these results, Assis et al. [89] suggest that this new formulation can be an alternative for topical application in skin wounds as a healing and antimicrobial agent.

Pickering emulsions are those stabilized by solid particles which possess higher biosafety and biocompatibility than classical emulsions stabilized by conventional surfactants. Pickering emulsions are able to spray and recover to high viscosity after spraying onto the wounds. High viscosity can be desired to provide a long residence time and intimate contact with the wounds [90]. A sprayed Pickering emulsion stabilized by chitosan nanoparticles was developed and the TTO was used as its oil phase. Curcumin was added to the oil phase to enhance the antioxidant and anti-inflammatory effects of the emulsion. Pickering emulsions were characterized and their antibacterial activities were evaluated, and the wound healing test was also performed. The authors also formulated a classical emulsion for comparing the results of both emulsion types. The classical emulsion also had the surfactants Tween^®^ 80 and Span^®^ 85. After injecting Balb/c mice, 6–8 weeks of age, with the mixed bacteria suspension subcutaneously, the wounds of the group treated with chitosan nanoparticles and the blank group presented purulence in contrast to the wounds treated with the Pickering emulsion. Moreover, it was observed that the wounds treated with the Pickering emulsion could heal normally and had the smallest area on the fifth day, suggesting that the Pickering emulsion displayed an excellent killing ability to bacteria avoiding wound infections. The wound healing rate in percentage and on day 10 was significantly higher (95.06%) in the group treated with the Pickering emulsion than those treated with classical emulsion (82.93%), chitosan nanoparticles (80.28%), and TTO (84.31%). Those results can be partially explained by the synergistic effects of TTO, curcumin, and chitosan nanoparticles. TTO had antimicrobial activity, anti-inflammatory properties, and scar prevention effects, while curcumin had great antioxidant and anti-inflammatory effects reducing the production of ROS during the inflammatory phase and promoting wound repair [90]. According to these results, the authors suggested that Pickering emulsion is a promising candidate for sprayed wound dressings.

Liposomes are phospholipid bilayer (unilamellar) and/or a concentric series of multiple (multilamellar) vesicles with a large aqueous inner core. They are constituted by synthetic and/or natural phospholipids (soybean lecithin, egg yolk, sunflower) and other membrane components (cholesterol). The size of liposomes ranges from 20 nm (nanoliposomes) to the micrometer scale with the phospholipid bilayer being 4–5 nm thick. Liposomes can encapsulate hydrophobic and hydrophilic drugs in their structure, being effective in drug delivery [91,92,93]. Aguilar-Pérez et al. [93] formulated and characterized nanoliposomes comprising various TTO concentrations (1.2–6.2 mM) and tested the antifungal activity against *Trichophyton rubrum* through the mycelial growth inhibition test. The same procedure was carried out for clove essential oil. Soybean and cholesterol were used as lipids, and Tween^®^ 80 as surfactant. The mycelial growth inhibition results of essential oil-loaded nanoliposomes were compared with pure EOs. The concentrations used for the assays were 0.25, 0.5, 1.0, and 1.5 μL/mL. The maximum encapsulation efficacy was observed for clove essential oil-loaded nanoliposomes. Nanoliposomes’ anti-fungal potential demonstrates their capability to inhibit mycelial growth at lower concentrations than the pure EOs against *T. rubrum* [93].

TTO-loaded nanoliposomes were also fabricated by Ge and Ge [94] using soybean phosphatidylcholine, cholesterol, and Tween^®^ 80. They characterized them and the antimicrobial activity was also assayed against *Staphylococcus aureus* ATCC 6538, *Escherichia coli* ATCC 8739, and *Candida albicans* ATCC 10231. Liposomes encapsulated TTO to form a stable liposome suspension, and the TTO-loaded nanoliposomes showed a significant increase in antimicrobial activity after encapsulation.

Liposomes may be part of nanofibers such as, for example, chitosan/poly(ethylene oxide) nanofiber mats containing TTO liposomes fabricated by using an electrospinning process [95]. These nanofibers had long-term and better antimicrobial activity against *S. aureus*, *E. coli,* and *C. albicans* than chitosan/poly(ethylene oxide) nanofiber mats. The combination of TTO-loaded liposomes and chitosan nanofiber mats act synergistically destroying the cell membrane, preventing cell adhesion, and causing the irregular aggregation of cytoplasm, according to the transmission electron microscope observation [95].

Ethosomes’ composition is based on phospholipids (such as soybean phosphatidylcholine), water, and ethanol content comprised between 20 and 45% *v/v* [96]. Similar to liposomes, ethosomes are able to solubilize both lipophilic and hydrophilic drugs inside the vesicles; nevertheless, ethanol has a higher loading capacity for lipophilic drugs than liposomes [96]. Azelaic acid, a dicarboxylic acid analog inhibits follicular keratinization and can be used topically to treat mild to moderate acne [97] Bisht et al. [97] fabricated ethosomes with azelaic acid and TTO to achieve a system with synergistic anti-acne properties within carbopol hydrogel. This hydrogel would make an easier application, adhesion, and drug penetration. The physical characterization was made and the authors also tested antibacterial activity against *S. aureus*, *S. epidermidis,* and *C. acnes.* The developed optimized ethanolic vesicle formulation of azelaic acid and TTO was compared with the topical marketed formulation in the testosterone-induced acne model in Swiss Albino mice [97]. The new hydrogel formulation with azelaic acid and TTO within ethosomes had significantly lower MIC values than TTO alone, maybe to the synergistic effect of azelaic acid and TTO, associated with the improved contact with the bacteria cell walls, increased contact time, and sustained drug delivery of the hydrogel [97]. Moreover, the new formulation-treated animals had a significant decrease in lesions, sebaceous gland hyperplasia, and seborrhea induced by testosterone in Swiss Albino mice [97].

Nanocapsules are polymeric nanocarriers prepared with poly ε-caprolactone, which is biocompatible, and biodegradable, of low toxicity, high stability, and low cost. The surface of nanocapsules can be modified to develop delivery systems able to interact more specifically with biological targets and enhance the intended biological activity [98]. In the pharmaceutical area, the modification has been conducted by using chitosan owing to its biocompatibility and biodegradability, and low toxicity [98]. Silva et al. [98] fabricated TTO-loading poly ε-caprolactone nanocapsules coated by chitosan for the topical acne treatment acting against anti-*C. acnes*. Chitosan with bioadhesive capacity would favor the drug retention in the skin surface and the positive charge would also contribute to the antimicrobial activity in combination with TTO, which would act synergistically. This hypothesis was confirmed by the authors after their experiment in which the coating of TTO-nanocapsules with chitosan presented higher anti-*C. acnes* activity (MIC = 0.14%, *v/v*) than the pure TTO (MIC = 0.56%, *v/v*) [98]. The poly ε-caprolactone nanocapsules did not present antimicrobial activity [98].

TTO has been also used for hand hygiene. For example, Youn et al. [99] compared the hand disinfection effects of TTO (5 mL of 10% tea tree oil disinfectant mixed in a ratio of 2:2:1:15 of *M. alternifolia* oil, solubilizer, glycerin, and sterile distilled water) with alcohol (2 mL of a gel-type hand sanitizer comprising 83% ethanol used without water), and benzalkonium chloride group receiving 0.8 mL of a foam-type hand sanitizer containing benzalkonium chloride used without water, and a control group with no treatment. The results were followed through subjective skin condition, transepidermal water loss, adenosine triphosphate, and a microbial culture test. The TTO group showed a remarkably higher disinfection effect, whereas the benzalkonium chloride group exhibited no disinfection effect based on adenosine triphosphate measurements. The control group demonstrated similar results to the benzalkonium chloride group. In view of the results, Youn et al. [99] suggest that TTO disinfectants should be introduced to nursing practice to prevent and reduce healthcare-associated infections.

The activity of different concentrations of TTO in diverse TTO-containing products such as a hygienic skin wash (HSW), an alcoholic hygienic skin wash (AHSW), and an alcoholic hand rub (AHR) was investigated against *S. aureus*, *Acinetobacter baumannii*, *E. coli* and *P. aeruginosa* [100]. The activity of the same formulations without TTO was used as a control. The efficacy of TTO was dependent on the formulation and the concentration tested, the concentration of interfering substances and the organism tested.

A complex based on natural antimicrobial and anti-irritant compounds, constituted by TTO (0.3%), eucalyptol (0.15%), α-bisabolol (0.10%), and silver citrate (0.01%) in specific ratio 30:15:10:1, respectively was mixed with soap base ingredients and tested for skin hygiene [101]. The antimicrobial activity was determined against the following microorganisms: *Bacillus cereus* ATCC 10702, *S. epidermidis* ATCC 14990, *S. aureus* ATCC 6538-P or ATCC 29213, *E. coli* ATCC 25922 and *P. aeruginosa* ATCC 9027, *Micrococcus luteus* 10240a, and *C. albicans* ATCC 10231. The combination displayed additive or synergistic activities against most strains studied, and a balanced performance between antimicrobial activity and biological safety, without skin irritant potential [101].

The antimicrobial activity of lavender, TTO, and lemon in washing liquid (1% alone or in mixtures) and O/W soft body balm (0.5% alone), was evaluated as well as combined with the synthetic preservative 1,3-dimethylol-5,5-dimethylhydantoin and 3-iodo-2-propynyl butyl carbamate mixture (0.1 and 0.3%) against *S. aureus* ATCC 6538, *P. aeruginosa* ATCC 9027, *Candida* sp. ŁOCK 0008 and *A. niger* ATCC 16404 [102]. In soft body balm formulations, oils at a concentration of 0.5% did not present any activity. The introduction of a solubilizer (polysorbate 80) to a system containing 0.5% TTO induced a substantial increase in bacteriostatic activity. A combination of 0.5% TTO, 5% solubilizer (polysorbate 80), and 0.3% synthetic preservative warranted the microbiological stability of soft body balm [102].

In antimicrobial terms, the comparison of three soaps for hand hygiene: 2.0% TTO, 0.5% triclosan, and 2.0% chlorhexidine was evaluated along with the perception of healthcare professionals about TTO [98]. For this, a determination of the logarithmic reduction of *E. coli* K12 colony-forming units before and after the hand hygiene of 15 volunteers was done, and interviews with 23 health professionals were performed. All the soaps demonstrated antimicrobial activity (a log_10_ reduction factor of 4.18 for TTO, 4.31 for triclosan, 3.89 for chlorhexidine, and 3.17 for reference soap), nevertheless, the TTO soap had the advantage to present a pleasant aroma and did not cause skin dryness [103].

Sgorbini et al. [104] investigated the permeation and release kinetics of the main constituents (terpinen-4-ol, α-terpineol, and 1,8-cineole) of TTO at different percentages (5–30% *w/w*) from several semisolid formulations (creams, ointments, and gels). The gel formulation had the highest percentages of those monoterpenoids’ release and permeation than creams and ointments, even at lower TTO concentrations, which means the possibility of using less concentrated formulations. By decreasing order, terpinen-4-ol, α-terpineol and 1,8-cineole were the most released and permeated compounds. In all cases, their skin retention was negligible, below 0.1% of the total amount in the formulation [104].

## 5. Anti-Inflammatory Properties of TTO and/or Their Fractions

Acute and chronic inflammatory diseases origin tissue damage due to the overproduction of pro-inflammatory cytokines, and other inflammatory mediators [IL-1, IL-6, TNF-α, nitric oxide (NO) synthesized by inducible NO synthase (iNOs), and prostaglandin E2 (Pg E2) synthesized by cyclooxygenase-2 (COX-2)], and also due to the action of nuclear factor-κB (NF-κB) that regulate pro-inflammatory gene’s expression during inflammation [105]. The heterodimers of NF-κB (predominantly p65:p50) are maintained in the cytoplasm of cells complexed to inhibitors of κB protein (IκB). Upon activation of NF-κB, there is a phosphorylation of IκB by IκB kinases (IKK), which allows a quick translocation of NF-κB into the nucleus that activates specific target gene expression, such as IL-1β, IL-6, TNF-α, and inducible enzymes (iNOS and COX-2) [105,106]. iNOS is mainly triggered and regulated by NF-κB transcription factor and mitogen-activated protein kinases (MAPKs) [106]. NF-κB and MAPKs are potential targets in therapies for several inflammatory symptoms [106].

In the review articles, some authors [105,106] concluded that terpenes are able to reduce the expression of IL-1β, IL-6, and TNF-α, either in specific macrophages cell lines or in vivo models such as Swiss mice and other rodents, although other transcription factors can be involved. Examples of monoterpenes with those abilities include limonene, α-phellandrene, linalool, terpinolene, borneol, myrcene, α-pinene, and terpinen-4-ol [105,106].

The metabolization of sebum lipids into free fatty acids by *C. acnes* triggers an inflammatory response via IL-1β release by monocytes and promotes perifollicular inflammation. In acne lesions, there are also higher levels of IL-17 than in healthy skin, nonetheless, higher levels were also found in the serum of patients suffering from acne. *C. acnes* is a powerful inducer of T helper 17 (Th17) lymphocytes which trigger the production of IL-17 [107]. IL-17 cytokines (IL-17A–IL-17F) bind to cytokine receptors (IL-17Rs), which are expressed in diverse cell populations, such as keratinocytes, fibroblasts, mesothelial cells, epithelial cells, and leukocytes [108]. IL-17A is the most studied member of the IL-17 family. IL-17A can interact with diverse mediators, such as interferon (IFN)-γ, IL-22, IL-1β, TNF-α, granulocyte-macrophage colony-stimulating factor (GM-CSF) to exert its proinflammatory effect [108].

IL-17, IL-21, IL-22, and Motif Chemokine Ligand 20 (CCL20) are released by Th17 cells. *C. acnes* promotes mixed Th17/Th1 responses by inducing the concomitant secretion of IL-17A and IFN-γ from specific CD4+ T cells [107]. Skin from acneic patients expressed high levels of pro-inflammatory cytokines IL-1α and IL-2) and low levels of anti-inflammatory cytokines such as IL-10 and IL-4 [107].

In vivo studies, particularly on imiquimod (IMQ) -induced psoriasis-like lesions in BALB/c mice, EOs and some of their constituents (e.g., *Lavandula angustifolia* Mill and *Perilla frutescens* (L.) Britton EOs, linalool, linalyl acetate) were able to inhibit the expression of the inflammatory factors IL-1, IL-6, iNOs, and COX2. At the same time, they decreased the expression of cytokines such as IL-6, IL-1, IL-23, IL-17, and NF-κB in distinct strengths [109,110].

The EOs isolated from *M. alternifolia* or their main constituents, such as terpinen-4-ol, have been demonstrated to be able to inhibit some pro-inflammatory cytokines by decreasing their expression in diverse cell lines, although very few have targeted skin inflammation. This can be observed in review articles in which the anti-inflammatory activity of TTO is reported but without references that support its application in dermatology [30,111,112]. 

Terpinen-4-ol-rich *M. alternifolia* essential oil (60–64%) in which the toxic hydrophobic monoterpenes (α-pinene, sabinene, α-terpinene, limonene), and the sesquiterpenes ledene and viridiflorol were previously removed, had the capacity to inhibit the cytokine production by the myeloid-derived cells when induced by lipopolysaccharide (LPS). LPS is an activator of NF-κB signaling in myeloid cell lines or macrophages and pro-inflammatory cytokine production. In myeloid-derived cell lines the terpinen-4-ol-enriched TTO, in the presence of the LPS activator, inhibited IκB phosphorylation and NF-κB signaling and translocation, resulting in the inhibition of iNOS protein expression and NO production. Moreover, murine RAW264.7 co-treated with the essential oil in combination with LPS markedly and dose-dependently inhibited the cytokine production of IL-3, IL-10, GM-CSF, and IFN-γ. For the human THP1 myeloid leukemia cell lines, terpinen-4-ol-enriched TTO dose-dependently inhibited the LPS-induced production of IL-1β, IL-6, and IL-10 [113]. 

In other cell lines, such as human peripheral blood monocytes, in the presence of LPS, the water-soluble components of TTO at a concentration of 0.016% (*v/v*) and predominantly constituted by terpinen-4-ol (>80%), were able to significantly reduce the production of TNF-α, IL-1 β, and IL-10 (by approximately 50%) and prostaglandin E2 (PGE2) (by approximately 30%) after 40 h [114]. Using the same protocol, but testing some individual components of TTO (terpinen-4-ol, α-terpineol, 1,8-cineole), the same authors reported that only terpinen-4-ol was able to suppress IL-1β, IL-8, IL-10, and PGE2. According to these results, the authors suggest that TTO and, particularly, terpinen-4-ol may control inflammatory responses in the skin, with the advantage of this compound may penetrate the vascularized dermis to exert anti-inflammatory activity. So, it would be necessary to select rich-terpinen-4-ol clones of *M. alternifolia* for further propagation.

The ability of TTO, terpinen-4-ol, and α-terpineol, to modulate the human monocytic cell line (U937) differentiated into macrophages response to bacterial LPS stimulation was studied by Nogueira et al. [115]. The results obtained showed that TTO and the individual volatiles reduced the production of IL-1β, IL-6, and IL-10, regardless of the origin of LPS (*Porphyromonas gingivalis* or *E. coli*), nevertheless, TTO decreased *E. coli* LPS-induced activation of NF-kB, p38 and extracellular signal-regulated kinase (ERK) MAPKinases but not *P. gingivalis*-induced activation. Terpinen-4-ol only reduced ERK activation induced by both *E. coli* and *P. gingivalis* LPS [115].

In vivo studies using BALB/c mice (20–25 g), which were previously subjected to terpinen-4-ol (5, 10, 20 mg/kg), given intraperitoneally, and 1 h after a solution of LPS (10 μg dissolved in 50 μL PBS) through intratracheal instillation, showed that monoterpenoid significantly inhibited LPS-induced inflammatory cytokines (TNF-α and IL-1β) through the suppression of phosphorylation of NF-κB p65 and IκBα [116] in the bronchoalveolar lavage fluid (BALF) of LPS-induced acute lung injury (ALI) mice. PPAR-γ, a nuclear receptor, is essential in glucose metabolism and cellular differentiation. It is also involved in the regulation of inflammatory response, in which activation of PPAR-γ can attenuate NF-κB activation. Terpinen-4-ol dose-dependently increased the expression of PPAR-γ. Ning et al. [116] also reported that terpinen-4-ol was able to suppress the activity of myeloperoxidase (MPO) in a dose-dependent manner, meaning an attenuation of inflammatory cell infiltrations. 

*Trypanosoma evansi* is a flagellate parasite that is transmitted by blood-sucking insects and vampire bats, parasitizing various species of domestic and wild animals. The disease caused by the parasite (trypanosomosis) originates from an increase in the levels of immunoglobulins (IgA, IgM, IgE, and IgG) and pro-inflammatory cytokines (TNF-α, INF-γ, IL-1, IL-6, IL-4, and IL-10) in mice experimentally infected [117]. These authors observed that TTO induced a decrease of IgM, IgA, and IgE in the male rats previously infected by *T. evansi* and an increase in IgG when compared to the control animals. TTO was able to reduce the levels of TNF-α, INF-γ, IL-1, IL-4, and IL-6, and increase the levels of IL-10 in healthy animals, nevertheless, in the presence of the parasite, the pro-inflammatory cytokines increased when compared to healthy animals, but was lower than the rats from the group not subjected to the TTO treatment [117]. Owing to these results along with the absence of hepatic and renal toxicity, and according to the biochemical parameters evaluated by Baldissera et al. [117], TTO can increase the longevity of animals infected by *T. evansi*. Later on, Baldissera et al. [118] also studied the effect of TTO on the innate immune response in silver catfish infected with the Gram-negative bacteria *Aeromonas hydrophila*. This microorganism is responsible for the epizootic ulcerative syndrome observed in silver catfish. The results also showed a decrease in the levels of the pro-inflammatory cytokines TNF-α, INF-γ, IL-1, and IL-6 in the infected animals previously treated with TTO, and an increase in the levels of IL-10 [118], as reported for rats previously infected by *T. evansi* [117]. At the same time, previous treatment of experimentally infected catfish with TTO was also able to prevent the increase of nucleoside triphosphate diphosphohydrolase (NTPDase), using adenosine triphosphate (ATP) or adenosine diphosphate (ADP) as substrate, and 5′-nucleotidase activities and to prevent the decrease on adenosine deaminase activity. However, it should be noted that the chemical composition of TTO in the two assays was different. In the first case [117], the percentages of terpinen-4-ol and γ-terpinene were 42 and 20%, respectively, whereas in the second work [118], the content of both compounds were practically the same (27 and 24%, respectively).

The production of *Macrobrachium rosenbergii* (de Man, 1879) (giant freshwater prawn) has expanded quickly due to its short breeding cycle, fast growth, and high nutritional value. Still, the larvae of this species undergo disease infestation readily. Antibiotics have been used to prevent this infestation, despite the risk of antibiotic resistance. For this reason, it is urgent to find alternatives safer and more effective, which can replace the utilization of antibiotics [119]. According to the authors, TTO and its main component terpinen-4-ol with antibacterial, antioxidant, and anti-inflammatory could be a solution. So, they design a study to explore the effects of dietary TTO levels and feeding patterns on the growth performance, antioxidant capacity, and intestinal immunity of *M. rosenbergii*. Their results showed that the supplementation of TTO (100 mg/kg) improved the growth performance of prawns when compared to the control group. However, the pro-inflammatory cytokines TNF-α, IL-6, and IFN-γ levels were not significantly different, only IL-1 decreased in the treated group. Moreover, the prolonged treatment (8 weeks with the TTO supplementation) induced an increase in TNF-α and IL-1 [119].

In a mouse model of acute colitis induced by heat stress (40 °C per day for 4 h) exposed for 14 consecutive days showed that the oral administration of terpinen-4-ol (5, 10, or 20 mg/kg) the production of TNF-α, IL-10, TLR4, and p65 was suppressed on day 1, 7, and 14 of heat stress. Histomorphological examination found that the groups submitted to the terpinen-4-ol presented better results than the heat stress group, whereby the authors concluded that this monoterpenoid had a protective effect on colonic tissue damage induced by heat stress regardless of the concentration used [120].

Blepharitis is a chronic inflammatory condition of the eyelids. The symptoms often persist despite treatment, but routine hygiene of the eyelid determines the therapeutic efficacy of the drug therapy [121]. These authors, in one group of patients, used a gel containing 3%-TTO, and in the other group, the gel used had TTO plus calendula oil, borage oil, vitamin E, and vitamin B_5_. In both groups, beyond the TNF-α, IL-6, and IL-1β levels evaluated at the first visit and 1 month after treatment, other parameters were evaluated such as Ocular Surface Disease Index (OSDI), tear breakup time (TBUT), ocular surface staining pattern, Schirmer’s test, impression cytology, and Demodex presence. According to the results obtained, the authors concluded that both formulations induced a decrease of tear cytokines and Demodex count, but were more significant in that gel with TTO, other oils, and vitamins. Moreover, IL-1β levels declined only by the treatment with the combined gel formula, but patients treated with TTO washing gel showed no significant decrease in the level of IL-1β [121].

There is also evidence that TTO can reduce histamine-induced skin inflammation through the intradermal injection of histamine diphosphate (5 μg in 50 μL), according to Kho et al. [122] who, for the first time, assayed the application of TTO (25 μL of pure oil) on injected histamine-induced weal and flare in the arms of more than twenty volunteers. The results showed that the mean weal volume significantly decreased 10 min after TTO application when compared to the control group in which liquid paraffin had been applied.

In the brief review carried out in the present work, it became clear that very few research works have focused on the effect of TTO on the inflammatory processes that are involved in acne vulgaris. From the results obtained in other types of cell lines or animals, the hypothesis of the potential of TTO as an anti-inflammatory in acne can be placed, however, there is a need for well-conducted and in-depth studies to corroborate or not the positive role that TTO may have in the acne treatment.

## 6. Antioxidant Properties of TTO

The ROS, produced by monocytes/macrophages and polymorphonuclear neutrophils (PMN) during stress, contribute to the host defense system against invading microorganisms. However, altered redox regulation of cellular signaling pathways may occur leading to an uncontrollable production and release of ROS, causing tissue damage [32,123]. The production of superoxide and nitric oxide, examples of ROS, by neutrophils and macrophages aids in phagocytosis and helps these cells in destroying bacteria. Nevertheless, superoxide anion radicals for killing any infecting bacteria also destroy the neutrophils and perhaps also damage surrounding tissue cells, contributing to the inflammation reaction [123]. In what concerns the TTO, it can act as antimicrobial by directly stimulating ROS production by PMN and increasing the intracellular ROS produced by monocytes, and simultaneously may act as an antioxidant and radical scavenger, by preventing the excessive production of ROS according to Caldefie-Chézet et al. [32].

According to some authors [123,124], diverse mechanisms can be found in antioxidant defense: inhibition or delay of the production of free radicals; free radical scavenging; modification of free radicals into less toxic compounds, since new radicals are formed during the scavenging process; delay or interruption of the formation of secondary toxic active species and mediators of inflammation; interruption of the chain propagation reaction; initiation and enhancing the endogenous antioxidant defense system; and chelating metal ions. Since several antioxidant mechanisms are involved, it is expected the existence of distinct methods for evaluating this capacity of samples.

Humans and other mammalian organisms have specific enzymatic and non-enzymatic antioxidants, which may act synergistically to protect cells against ROS damage. Examples of endogenous antioxidant enzymes include glutathione peroxidase (Gpx), catalase (CAT), and superoxide dismutase (SOD); and examples of non-enzymatic endogenous antioxidants are glutathione, melatonin, or Coenzyme Q_10_, among other examples [124]. Nevertheless, endogenous antioxidants may not be enough to prevent oxidative stress and, to keep cellular functions, exogenous antioxidants may be included (e.g., vitamin E, vitamin C, phenols, EOs) [114,124].

### 6.1. Antioxidant Properties of TTO in Food Matrices

The antioxidant activity of TTO has been evaluated in in vitro and in vivo models. Vo et al. [125] wanted to enhance the extraction performance of TTO by hydrodistillation by adding surfactants (nonionic Triton CG-110). An optimal extraction yield of TTO was predicted by the response surface methodology (RSM) under the following conditions: 597 mg/L Triton CG-110 as a surfactant; a ratio of liquid extractant/desiccated leaf at 25.4 mL/g; and 140 min for the extraction time. Using the same method (RSM), the maximal antioxidant activity should be in the following conditions: 689 mg/L Triton CG-110 as a surfactant; a ratio of liquid extractant/desiccated leaf at 24.1 mL/g; and 128 min for the extraction time. According to these data, the best conditions for obtaining the maximal antioxidant activity were different from for obtaining the maximal TTO yield, therefore, the extraction must be conducted depending on the final goal: better yield or better antioxidant activity [125]. At the end of the extraction, the surfactant nonionic Triton CG-110 does not remain in the TTO because it is much less volatile than TTO.

The antioxidant activity assays have been carried out to find their effects on the growth, quality, and preservation of diverse matrices (meat, fish, vegetables, fruits), on wound healing, and preventing skin photoaging as reported below. The effects of TTO-supplemented diets (25, 50, 100, 200, and 400 mg/kg) on the performance, intestinal antioxidant activity, and non-specific immunity of prawns after ammonia nitrogen stress were evaluated. The results showed that 100 mg/kg of TTO significantly increased the performance and survival rate of prawns. Concentrations of 100 and 200 mg/kg TTO significantly increased SOD activity (an enzyme that catalyzes the dismutation of superoxide radical anions to hydrogen peroxide and molecular oxygen to eradicate oxidative damage) and decreased the amounts of malondialdehyde (MDA), a product resulting from lipid peroxidation, in prawns’ intestines [126]. Under ammonia stress, the prawns fed with 100 mg/kg TTO had the highest survival rate, an improved antioxidant capacity by increasing the anti-superoxide anion activity and SOD, an upregulation of the intestinal relative expression of antioxidant-related genes (*peroxiredoxin-5*), TNF-α, and IL-1, and an increase of the levels of iNOS activities and NO contents [126]. The decrease in MDA level and the increase of Gpx in the serum of finishing pigs were also reported [127] when the animal was fed with TTO supplementation of 200 or 300 mg/kg. The supplementation with 200 mg/kg TTO also increased the Gpx activity in the *longissimus dorsi* muscle, accompanied by an improvement in growth performance and meat quality.

Due to the antioxidant and antimicrobial activities reported for TTO, it has been a target of study as a preservative in the postharvest of fruits and vegetables. In raspberries, TTO was spotted on a piece of filter paper which was subsequently hung inside the plastic containers just before the lids were closed. The volatile compounds were allowed to vaporize inside the containers spontaneously at 20 °C, for 16 h. Methyl jasmonate, another compound assayed, evidenced much better antioxidant activity than TTO [128]. Similar results were observed in another study, in which the methyl jasmonate (22.4 μL/L) treatment promoted the antioxidant activity in raspberries and blackberries as measured by the 2,2-diphenyl-1-picrylhydrazyl (DPPH) and 2,2′-azino-bis(3-ethylbenzothiazoline-6-sulfonic acid (ABTS) scavenging free radicals, as well as increased scavenging capacities on the superoxide radical, hydrogen peroxide, and singlet oxygen. Treatment with TTO (100 μL/L) improved these free-radical scavenging capacities, except for hydrogen peroxide in strawberries, superoxide radical, and singlet oxygen in blackberries [129].

In leafy vegetables, mainly in the Swiss chard, TTO at 0.09% (*v/v*) and 0.18% (*v/v*), reduced peroxidase activity in 29 and 42%, respectively. Being peroxidase, a plant enzyme associated with off-flavors in fruits and vegetables, this ability of TTO to reduce its activity is useful to preserve horticultural products’ quality and extend their shelf-life when minimally processed [130]. However, it is necessary to guarantee the absence of deleterious effects on the sensory attributes of vegetables, particularly on the aroma [131]. In Swiss chard, after 1 day and up to day 14 of storage, the panelists did not find differences in odor or other sensory indexes, but there was no enhancement of the sensory quality when compared to the control samples without TTO [131]. The reduction of peroxidase activity can be seen as very important in vegetables with high rates of browning since peroxidase, along with polyphenol oxidase, catalyze reactions that lead to the browning of foods and off-flavors and nutritional damage [130].

Edible films have been developed in order to replace the non-biodegradable synthetic packaging of vegetables and fruits. Chitosan is a polysaccharide derivative of chitin with antimicrobial, antioxidant, and UV-barrier [132]. This polymer has been tested as a film or coating in food which can be associated with other antimicrobial or antioxidant compounds, such as EOs (e.g., TTO) serving as their vehicles [133]. Chitosan film obtained in malic acid and supplemented with TTO showed higher antioxidant activity than the film without malic acid. This antioxidant activity was dose-dependent (0.5 and 1% TTO). When the film was made with chitosan and lactic acid, there was no enhancement of the antioxidant activity measured through the capacity of scavenging DPPH and ABTS free radicals. The application of chitosan film with TTO (0.5%) as a coating on fresh-cut green beans led to a significant reduction in the browning index and an increase of the antioxidant capacity up to 15 days postharvest [133].

Kim et al. [26] reported the strong antioxidant activity of TTO, even comparable to the synthetic butylated hydroxytoluene (BHT), using two methods: the DPPH assay and the hexanal/hexanoic acid assay. The activities were attributed by descending order to *α*-terpinene > α-terpinolene > γ-terpinene. However, since these components prevent the oxidation of DPPH or hexanal, such can be considered as those monoterpenes had undergone oxidation. If so, TTO can become unstable originating electrophilic and reactive species and, consequently, potential skin sensitizers [134,135]. Such must be deeply surveyed. The strong activity obtained by Kim et al. [26] contrasts with that reported by Borotová et al. [63], in which the same volume of sample (10 μL) had an activity of 80%, measured through the DPPH method, whereas Borotová et al. [63] only had 40% of activity. The different chemical composition of TTO was pointed out by these authors as a possible factor in the different percentages of antioxidant activity. The low antioxidant activity of TTO, using the same detection procedure, was also reported by Martiniaková et al. [136] when comparing several EOs. This work permitted the authors to conclude that Ceylon cinnamon had the highest activity followed by clove oil, therefore they could be used as natural alternatives to synthetic antioxidants in cosmetic products, nevertheless, some components present in these oils can induce allergies.

### 6.2. Antioxidant Properties of TTO in Non-Food Matrices

Pormohammad et al. [76] reported the ability of TTO to prevent the formation of intracellular H_2_O_2_ (12.5 relative fluorescence units (RFU and 18.5 RFU for untreated control), and superoxide anion radical (5 RFU and 16.7 RFU for untreated control) in *C. elegans* measured through the dichlorodihydrofluorescein diacetate (DCFH-DA) and dihydroethidium (DHE) probes, respectively. Reduced glutathione (GSH) a natural antioxidant was increased when *C. elegans* was submitted to TTO (729 RFU and 312.6 RFU for untreated control), measured using the naphthalene-2, 3-dicarboxaldehyde (NDA) probe.

The comparison of the antioxidant activity of TTO with other Australian EOs revealed that TTO (IC_50_ = 1.2 × 10^−2^ mL/mL, 3.5 × 10^−2^ mL/mL)had lower ability than Australian blue cypress oil (IC_50_ = 9.5 × 10^−3^ mL/mL, 3.0 × 10^−3^ mL/mL), lemon-scented ironbark oil (IC_50_ = 6.4 × 10^−3^ mL/mL, 7.0 × 10^−3^ mL/mL), lemon-scented eucalyptus oil (IC_50_ = 4.8 × 10^−3^ mL/mL, 8.9 × 10^−3^ mL/mL), and lemon-scented tea-tree oil (IC_50_ = 1.5 × 10^−3^ mL/mL, 1.5 × 10^−3^ mL/mL), evaluated through the DPPH assay and ABTS assay, respectively. TTO had much lower activity than the positive control the vitamin E [137].

For accelerating wound healing, several strategies have been developed and one of them consists in the utilization of antioxidants, since around the moist microenvironment of the wound occurs oxidative stress. With this purpose, Hu et al. [138] constructed a chitosan-based bioinspired asymmetric wound repair composite film, constituted by two different functional layers: a hydrophilic chitosan/silk fibroin repair layer, and a hydrophobic bacteriostatic TTO layer with a rough surface. At the same time, sodium ascorbate—entrapped poly (lactic-*co*-glycolic acid) microspheres were distributed homogeneously in the hydrophilic layer. The results showed that this film exhibited excellent biocompatibility, and good antibacterial capacity, enhancing not only the adhesion and proliferation of the fibroblast cell line that was isolated from a mouse NIH/Swiss embryo (NIH3T3) cell in vitro but also facilitating the healing of a rat full-thickness skin wounds model. Simultaneously, it was observed, through the DPPH method, that the film presented antioxidant activity after incubation for 12 h, which was ascribed to the release of sodium ascorbate and TTO [138]. According to the authors, this asymmetric film could be of great importance in wound healing and related soft tissue regeneration. Doustdar et al. [139] produced poly (ε-caprolactone)/soy protein isolate (PCL/SPI, 95:5) mats with different concentrations of TTO (6, 12, and 20%), using the electrospinning method. The release profiles showed that higher amounts of TTO in the mats could be released in an acidic environment and the antioxidant activity of the mats increased by the increase in their TTO content (PCL/SPI TTO 6%, PCL/SPI TTO 12%, PCL/SPI TTO 20%, the inhibitions percentages, measured through the DPPH method, were 30, 45, and 65%, respectively). The cell viability of NIH3T3, cell adhesion, and live/dead assay of TTO-loaded mats claimed that the mats were biocompatible, therefore, with great possibility for being used in wound dressings [139].

Several factors trigger skin photoaging such as ultraviolet radiation, infrared radiation, and visible light, which induce the increase in free radical production in skin. The utilization of antioxidants in cosmetic formulations has an important role in the prevention of free radical production. Malik and Upadhyay [140] aimed to determine the antioxidant activity, sun protection factor value, and half-maximal inhibitory concentration of rosemary essential oil and TTO. They concluded that both EOs had antioxidant activity evaluated through the DPPH and nitric oxide (NO) methods, but rosemary oil had a higher sun protection factor, measured through the UV spectroscopy method. According to the authors, rosemary and TTO could offer a synergistic sun protection factor effect, antioxidant activity (IC_50_ = 0.25 mg/mL for both methods), and anti-aging activity of cosmetic preparations. Infante et al. [141] evaluated the skin penetration of *M. alternifolia* nanoemulsion using confocal Raman microspectroscopy, and antioxidant properties by electron paramagnetic resonance spectroscopy for pure TTO as well for its nanoemulsion. In the in vivo studies, the authors used 40 male participants, aged 18–28 years. Before and after 90 days of study, skin hydrolipidic and morphological characteristics were studied. The nanoemulsion presented 10 times lower antioxidant activity than the pure TTO but higher penetration through the stratum corneum, which resulted in pronounced effects, that is, improving the epidermis morphology and the dermal echogenicity. The authors [141] concluded that TTO into nanoemulsions would be able to improve photoaged skin, reaching deeper skin layers.

Flores et al. [142] prepared hydrogels containing nanocapsules and nanoemulsions with TTO, to reduce the edema induced by UVB exposure. The authors observed that hydrogel containing nanocapsules with TTO presented a higher reduction of the wound area compared to the hydrogel containing nanoemulsions or hydrogel containing allantoin. To understand the healing process of the TTO within the formulations, Flores et al. [142] studied different antioxidant defenses in the rat lesions submitted to the treatment, such as reduced glutathione (GSH), ascorbic acid level, and catalase activity. The results showed higher GSH levels and CAT activity in those animals submitted to the action of hydrogel-containing nanocapsules with TTO than the remaining formulations. Such results could be explained by the polymeric wall against the TTO volatilization. Therefore, Flores et al. [142] believe that this type of formulation with TTO can be used as a potential treatment for inflammatory disorders in which antioxidant mechanisms are involved, and wound healing.

## 7. Toxicity of TTO

In spite of the utilization of TTO in the treatment of foot problems, particularly in tinea pedis and toenail onychomycosis, or the treatment of acne, dandruff, head lice, and recurrent herpes labialis, this EO is not free of adverse effects. The volatiles that constitute TTO are able to penetrate skin or increase penetration of other compounds and trigger toxicity [143,144]. Severe reactions can be considered extremely rare in the absence of ingestion, [145] although some components of TTO are responsible for some allergic reactions (1,8-cineole, terpinen-4-ol, limonene, among other ones), nevertheless, some authors believe that most responsible is the oxidized TTO and not fresh TTO, whereby it will be necessary to guarantee adequate storage conditions for preventing allergic reactions [146,147]. Recently, Avonto et al. [134] annotated that terpinolene, α-terpinene, and terpinen-4-ol of TTO were unstable originating the formation of electrophilic and reactive species in accelerated aging conditions, evaluated through high throughput screening method dansylcysteamine adducts (HTS-DCYA). Moreover, α-terpinene autoxidizes rapidly under air exposure to originate skin allergens. These reactive species are potential skin sensitizers [134,135]. The allergic contact dermatitis caused by TTO also depends on the product used. For example, the mix of TTO with other EOs (turpentine oil) or fragrances (colophonium, *Myroxylon pereirae* Royle) may exacerbate contact dermatitis [147,148]. These authors also recommend that concentrated forms of TTO (>10%) must be avoided particularly on damaged skin.

After ingestion, some adverse effects were reported by Carson and Riley [149]: oral toxicity in a 23-month-old boy presenting dermatitis after topical application; remarkable rash after ingestion of half a teaspoon of TTO by a 60-year-old man; and following ingestion of half a teacup of neat TTO induced coma for 12 h and semi-coma for a further 36 h. In another case, the ingestion of 2 teaspoons of 100% pure TTO by a 4-year-old boy led to symptoms of ataxia within 30 min followed by unconsciousness and unresponsiveness requiring intubation [150]. These authors also reported TTO toxicity in cats and dogs which were expressed through depression, weakness, incoordination, and muscle tremors. Concerning other animals, a one-year-old, 80 g male cockatiel (*Nymphicus hollandicus*) also presented serious depression episode with severe liver damage and slight renal involvement, and moderate neutrophilia moderate neutropenia, after application of 3 drops (0.15 mL) of 100% pure TTO on the cutis of its right wing and part of the thorax [151]. Shortly, regardless of the less complicated adverse effects described for TTO, more pronounced side effects cannot be discharged.

## 8. Efficacy of TTO in Acne Therapy—Studies in Humans

To learn about the efficacy of TTO in the treatment of acne in humans, a search on the available literature was conducted. We selected all complete original articles, written in English or Portuguese that described trials conducted in humans using TTO in pharmaceutical forms applied to the skin, in some study groups, and that presented efficacy data on its use. Parameters such as skin oiliness, the number of inflammatory and/or non-inflammatory lesions, and the degree or severity of acne were considered as data on product efficacy. 

Various studies have demonstrated the efficacy of TTO in several human pathologies, such as dentistry, infectious diseases, ophthalmology, and dermatology, amongst others [152]. In dermatology, and specifically, in acne vulgaris, several essential oils have shown beneficial results. Tea tree oil is one of the oils described in tests in vivo as having biological activity on acne, thanks to its antimicrobial, anti-inflammatory, and antioxidant properties [9].

Several human studies have been found evaluating the efficacy of TTO in topical preparations for acne vulgaris (Table 2) [153,154,155,156,157,158,159,160,161,162]. All of them have shown efficacy in the treatment of this pathology, namely in inflammatory lesions. In addition to this demonstrated efficacy, the low incidence of adverse effects described is also an advantage for the use of this oil in topical products for the treatment of acne vulgaris.

In 1990, a single-blind random clinical trial (RCT) was carried out on subjects with mild-moderate acne, for three months, comparing the use of a TTO (5%) water-based gel and benzoyl peroxide (BP) 5% water-based lotion. Although both groups showed a reduction in the inflammatory lesions, the group that applied BP showed a significantly greater improvement than the group that applied TTO (*p* < 0.001). The same was possible to observe in the reduction of skin oiliness. In non-inflammatory lesions, both groups showed a significant decrease and there were no differences between the groups. Although the results point towards greater efficacy of the BP in reducing acne lesions, the results regarding safety point to better results with the TTO gel. In this group, the adverse events reported were less (44%) than in the BP group (79%), with a statistically significant difference (*p* < 0.001) [153]. In fact, BP has well-described antibacterial, anti-inflammatory, keratolytic, and wound-healing properties, making it a good treatment for acne. Similarly, concentration-dependent adverse reactions are also known, such as skin irritation, irritant dermatitis with erythema, scaling, or itching. These reactions may limit the use of BP in some individuals [163].

Compared with the placebo (carbomer gel), the TTO 5% gel demonstrated good anti-inflammatory and antibacterial capacities, leading to a significant reduction in inflammatory lesions (papules 46.06%; pustules 47.45%) in a double-blind RCT developed by Enshaieh et al. [154], on subjects with mild-moderate acne, for 45 days. This RCT also showed a significant decrease in the number of comedones (40.24%), the total number of lesions (43.64%), and the acne severity index (ASI) (40.49%). TTO 5% proved to be 3.55 and 5.75 times more effective in reducing the total number of lesions and the severity of acne, respectively. Regarding safety, more adverse effects were reported in the TTO group compared to placebo, namely pruritus, burning sensation on application, and scaling. However, their incidence was low and topical application of TTO 5% gel may be considered tolerable [154].

The use of TTO in cleansing and moisturizing products for acne was evaluated in a dual-center, open-label, phase II pilot study, which assessed the efficacy and safety of two products containing TTO, a Face Wash (7 mg/g) and a Gel (200 mg/g), on mild-moderate acne vulgaris, for 12 weeks. The results showed a significant 54% decrease in the total number of lesions (*p* < 0.001), as well as in the investigator's global assessment score (*p* < 0.05). Furthermore, skin oiliness decreased significantly (*p* < 0.01). Even with a twice-daily application, no serious adverse reactions were reported. Similar to other studies, moderate scaling, peeling, and dryness were described. This study also evaluated the acceptability of the gel by the participants by means of a questionnaire at the end of the study. Results showed that the gel was well accepted with mean scores between 3.5 and 4.6 (highest score = 5). The cosmeticity of the product (texture, consistency, and ease of application) was the feature with the highest scores. Characteristics such as fragrance and absorption of the product were those that obtained lower score values [156]. TTO has a peppery-spicy, camphor-like aroma with a less intense cooling effect than menthol [1]. These characteristics could be beneficial in an anti-acne product as the sense of freshness may lead to greater acceptability, however, the more intense aroma may be the reason for the lower scores described above.

Efficacy and safety were also evaluated in a double-blind split-face RCT in subjects with mild-moderate acne vulgaris for 8 weeks. In this study, TTO was 5% as compared with *Lactobacillus*-fermented *Chamaecyparis obtusa* (LFCO). The results showed greater efficacy for the use of LFCO in reducing the number of inflammatory and non-inflammatory lesions, as well as in reducing sebaceous secretion and the size of the sebaceous glands [155]. In these last two aspects, the decrease was not significant for TTO, contrary to what was described by Malhi et al. [156] who described a significant decrease in skin oiliness. However, the anti-inflammatory properties of TTO were enhanced in this study with a significant decrease in inflammation-related proteins, namely IL-8 and TLR-2 mRNA. Adverse reactions such as dryness, erythema, and desquamation were once again described as the most prevalent in the use of topical TTO [155].

Adapalene is used in the treatment of acne vulgaris with good clinical results. It is a synthetic derivative of naphthoic acid with retinoid activity, i.e., it has keratolytic, anti-inflammatory, and antiseborrheic properties. These properties are essential in the treatment of acne, as they enable treatment to be targeted both to non-inflammatory lesions resulting from hyperkeratinization (comedones) and inflammatory lesions (papules) [164]. However, this compound does not have significant antibacterial properties and, therefore, its action is less at the level of inflammatory lesions such as pustules. The efficacy of the association of adapalene and TTO vs. adapalene was evaluated in a triple-blind RCT in subjects with mild-moderate acne vulgaris for 12 weeks. In the control group (n = 47) adapalene gel 0.1% was used, once daily at night, and in the intervention group, tea tree oil (6%) nanoemulsion containing adapalene gel 0.1% (TTO + ADA) was also used once daily. The results were very positive for the intervention group with a greater decrease in the number of inflammatory and non-inflammatory lesions and the number of total lesions. The ASI showed a more accentuated decrease in the intervention group with 71.69% of treatment success compared with the control group (6.38%) [158]. The association between adapalene and TTO makes it possible to obtain a topical formulation that covers the four main factors in the development of acne: hyperseborrhoea, hyperkeratinization, inflammation, and bacterial colonization. Despite the good results obtained, further studies should be carried out to obtain more consistent results that corroborate those already obtained in the work developed by Najafi-Taher et al. [158].

The anti-inflammatory and antibacterial properties of TTO have been described and are mainly due to terpinen-4-ol [111], being widely used and studied in the treatment of acne. However, this compound does not have excellent antiseborrheic and keratolytic properties, and it may therefore be necessary to associate it with other compounds that not only reinforce but also complement its activity in the treatment of acne, as described above. The search for phytotherapeutic ingredients has led to some studies evaluating the efficacy of formulations with various plant extracts. Of the ten human trials shown in Table 2, five used formulations containing TTO in addition to other herbal compounds with activity on acne [157,159,160,161,162].

An uncontrolled, open-label multicentric phase III RCT evaluated the efficacy of a topical gel with TTO 5% (group II) vs. oral administration of tablets containing Neem extract 200 mg, Turmeric extracts 200 mg and Piper extracts 10 mg (group I) vs. a combination of gel and tablets (group III), over a 4-week period, in mild-moderate acne. Group II (n = 46) showed improvement in non-inflammatory lesions (blackheads) (78.3%) with results such as the other groups under study. As regards the inflammatory lesions such as papules and nodules, group II showed improvements, but less than those described for the other groups. As for pustules and cysts, the results showed more significant improvement than in the other groups. In general, the topical application of TTO 5% showed a 17% improvement in deep inflammatory lesions compared to the oral administration of tablets and an improvement of more than 20% compared to group III (gel + tablets). Despite the improvements, the study concluded that the improvement results achieved in group II were slightly lower than those achieved in the other study groups [160]. The extracts used in the tablets have antibacterial and anti-inflammatory activity and can therefore reinforce the action of the TTO gel, thereby achieving better results. *Azadirachta indica* A. Juss. (Neem) the extract has antioxidant properties, and its application in cosmetic products and nutritional supplements for various skin problems, such as acne, has been widely studied [165]. *Curcuma longa* L. (Turmeric) extract has anti-inflammatory properties in addition to its antibacterial ones [166]. An open-label, single-center, single-arm, a four-week clinical study evaluated the efficacy and safety of an herbal skincare product in the prevention and/or reduction of mild-to-moderate acne (Purifying Neem Face Wash, a combination of extracts of neem and turmeric with anionic and amphoteric surfactants). The results showed a reduction of inflammatory and non-inflammatory lesions, as well as a reduction in sebum production and an improvement in skin hydration, indicating the beneficial effects of herbal ingredients in acne prevention and reduction [167]. *Piper nigrum* L. extract has the least scientific support in dealing with acne but has shown good antimicrobial activity [168,169].

Like TTO, lavender oil has antibacterial, antioxidant, and anti-inflammatory properties due to its wide variety of constituents such as mono and sesquiterpene and phenolic acids [170,171,172]. The effect of concomitant use of TTO (3%) and lavender oil (2%) on acne vulgaris was evaluated in an experimental pretest-posttest. The results showed significant results in the decrease of inflammatory lesions (*p* < 0.001) and the colonization of lesions by *C. acnes* (*p* = 0.005). The decrease in non-inflammatory lesions and the total number of lesions was significant in both groups, although more accentuated in the group that used the association of oils. This group also presented a significant reduction in sebum excretion rate (*p* = 0.004) while in the control group, this reduction was not significant (*p* = 0.112). Topical application of TTO and lavender oil seems to be safe, since only one individual (out of 27) reported itch 2 to 3 days after starting treatment, resolving quickly with no subsequent adverse effects [161].

A single-center, randomized, double-blinded, comparative study was designed to compare erythromycin 3% cream with a topical formulation (cream) containing TTO 3%, propolis extracts 20%, and aloe vera leaf juice 10%. This formulation demonstrated a significant reduction in the number of comedones, papules, and pustules (*p* < 0.001), as well as in ASI (*p* = 0.0368) and the total number of acne lions (*p* = 0.001). The topical formulation developed for the study was more effective in reducing erythema scars than erythromycin (*p* = 0.003) but there were no differences between groups in the reduction of popular erythema [159]. It cannot be ruled out that the formulation tested in this study contained other ingredients with therapeutic properties in acne treatment, described in the literature, namely propolis and aloe vera. Propolis, although its chemical composition in polyphenols and flavonoids may vary depending on its provenance, has antibacterial and anti-inflammatory properties that may be beneficial in the treatment of acne [173,174,175]. *Aloe vera* (L.) Burm.f., also has antibacterial, anti-inflammatory, and antioxidant properties as well as moisturizing properties that appear to be beneficial in treating acne [176,177,178,179,180].

An observer-blinded, noninferiority randomized controlled study evaluated the efficacy of a topical gel formulation of various plant extracts (onion, Lavandula, mangosteen, aloe vera, paper mulberry, and tea tree extracts) versus BP cream 2.5%, in the treatment of mild-moderate acne. Although the results were positive in reducing the number of comedones, inflammatory lesions, and the total number of acne lesions in the intervention group (plant extracts), the results in the control group (BP 2.5%) were more marked. The mean difference between groups in the reduction in comedones, inflammatory lesions, and total lesion count at the end of the study showed an inconclusive result for noninferiority [157]. Onion (*Allium cepa* L.) presents limited scientific evidence on its use in acne. Keratolytic, antimicrobial, and anti-inflammatory properties have been described, which may be beneficial in the treatment of acne. However, the unpleasant odor may limit its use in products for topical application, adding to the few results published with this extract. Mangosteen (*Garcinia x mangostana* L.), rich in xanthone compounds, has antibacterial and anti-inflammatory properties, showing positive results in the treatment of acne [181,182,183]. Paper mulberry (*Broussonetia papyrifera* (L.) L’Hér. Ex Vent.) contains several chemical compounds such as flavonoids, polyphenols, alkaloids, coumarins, and saponins, showing a wide range of biological effects, including antibacterial, anti-inflammatory and antioxidant action, which may be beneficial in the treatment of acne [184,185,186]. In addition, it has a high tyrosinase inhibitory effect and can be used as a bleaching ingredient. Although the non-inferiority test did not assess this outcome, this ingredient may be beneficial in reducing post-inflammatory hyperpigmentation in acne [186]. Regarding safety, no systemic adverse effects were reported from the use of the plant extract gel, and the most reported effects were skin irritation, itching, and burning sensation after application. One subject from the intervention group withdrew from the study due to an increase in acne lesions, and three (n = 3) subjects reported itching and small pink papules on the treatment areas that disappeared after discontinuing the use of the gel for 4 days without using topical steroids and did not experience these adverse effects for the remainder of the study. Other outcomes were also assessed, namely treatment adherence and subjects’ satisfaction with the product. In these parameters, both adherence (*p* = 0.002) and satisfaction with the administration of the product (*p* = 0.001) were significantly higher for the intervention group compared with the control group [157].

The most recent human study found in the literature did not have as its main objective the evaluation of the efficacy of TTO in the treatment of acne but to assess the feasibility of in vivo reflectance confocal microscopy as a method to evaluate the efficacy of the joint use of a cleansing gel (Ivapur purifying cleansing gel) and a cream (Ivapur K cream). Among other constituents, Ivapur K cream contained TTO. The results showed a significant reduction in skin oiliness (*p* < 0.001) and follicular colonization by *C. acnes* (*p* = 0.003), as well as in the number of comedones (*p* < 0.001) and the density of the inflammatory infiltrate (*p* < 0.001). It was also possible to observe an increase in the number of regular follicles (*p* < 0.001) after using both products for a 28-day period in subjects with oily, acne-prone skin [162]. Although TTO may have contributed to the results obtained through synergy with the other ingredients, it is important to note that Ivapur purifying cleansing gel includes zincidone^®^ (INCI: Zinc PCA), the zinc salt of L-Pyrrolidone Carboxylic acid, which has seborregulator properties [187,188]. Zinc compounds are also known for their anti-inflammatory and antimicrobial properties, among others, and are widely used in acne treatment products [189,190,191]. Ivapur K cream was the formulation that contained TTO, but also other ingredients such as hydroxy acids (lactic acid and salicylic acid), which due to their keratolytic properties show good results in the treatment of acne [192,193,194], piroctone olamine which has good antimicrobial and essentially antifungal properties [195,196,197], and bisabolol known especially for its anti-inflammatory and antimicrobial properties [112,198,199,200]. The beneficial results described in the study [139] may thus come from the synergy of the various ingredients in the approach to the various pathological aspects of acne.

In general, the studies found in the literature demonstrate the good properties of TTO in treating acne, both in isolation and associated with other ingredients with complementary activities. Of the 10 studies described, 5 used TTO in isolation and the remainder used it in association with other extracts. Only one RCT evaluated the efficacy of TTO in relation to placebo and two RCTs evaluated its efficacy in relation to other compounds with known activity in the treatment of acne, such as BP. The number of individuals included in the studies varied, there is one study with 14 individuals [156] and one with 53 individuals [158]. All involved young people with mild-moderate acne, except the study by Lupu et al. [162] which assessed the use of topical products on oily, acne-prone skin. However, the main objective of this study was not to assess the efficacy of the products, but rather a methodology that would allow the assessment of alterations resulting from the use of anti-acne products. However, given the heterogeneity of the studies found, it is difficult to obtain clear results and conclusions, as described in two reviews [152,201].

## 9. Conclusions

The traditional uses of tea tree oil include treatment of small superficial wounds and insect bites; small boils (furuncles and mild acne); relief of itching and irritation in cases of mild athlete’s foot; minor inflammation of the oral mucosa; that is, these indications expressed in the European Union Herbal Monograph on *Melaleuca alternifolia* [202], are exclusively based on long-standing use. After the compilation made in the present review shows that almost all studies on the biological properties are in vitro. Even so, in many cases, the authors are not able to establish a correlation between the chemical composition and the biological properties observed. In other studies, the authors attribute the TTO activities to the synergism effect among the constituents of the essential oil or mixtures of essential oils of diverse origins. Despite the studies focused on the mechanisms involved in the anti-inflammatory and antioxidant activities of TTO in diverse systems and matrices, very few reported such effects in the acne vulgaris, notwithstanding the traditional use including mild acne. The clinical studies also show heterogenic results since the authors utilized in their studies diverse types of formulations added or not with other active constituents. In addition, the study objectives are also diverse (effect on comedones, pustules, papules, cysts, acne severity index, total lesion count, facial oiliness, *C. acnes* colonization, anti-inflammatory). This diversity of results along with the very few consistent clinical studies make it difficult to find a relationship between TTO and anti-acne effect through anti-inflammatory and antioxidant attributes.

More robust studies need to be carried out with common procedures, formulations, and objectives, in several accredited laboratories in different parts of the globe, using groups of individuals with well-defined acne. Without this type of approach, many studies will continue to be carried out, without resulting in a practical result, that is, only conclusions of academic interest will result.

## Figures and Tables

**Figure 1 antioxidants-12-01264-f001:**
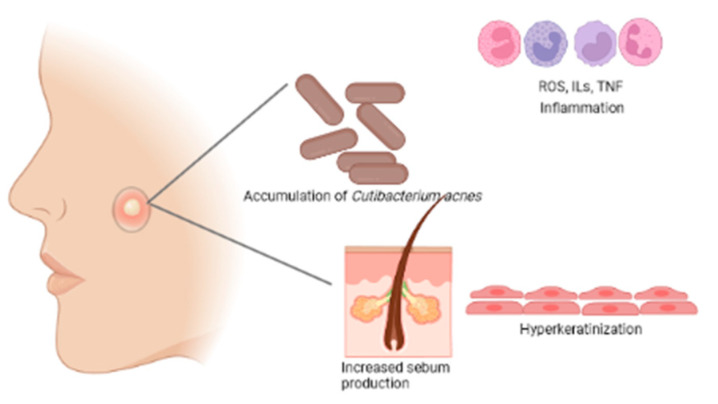
Schema of acne vulgaris (Created with BioRend).

**Figure 2 antioxidants-12-01264-f002:**
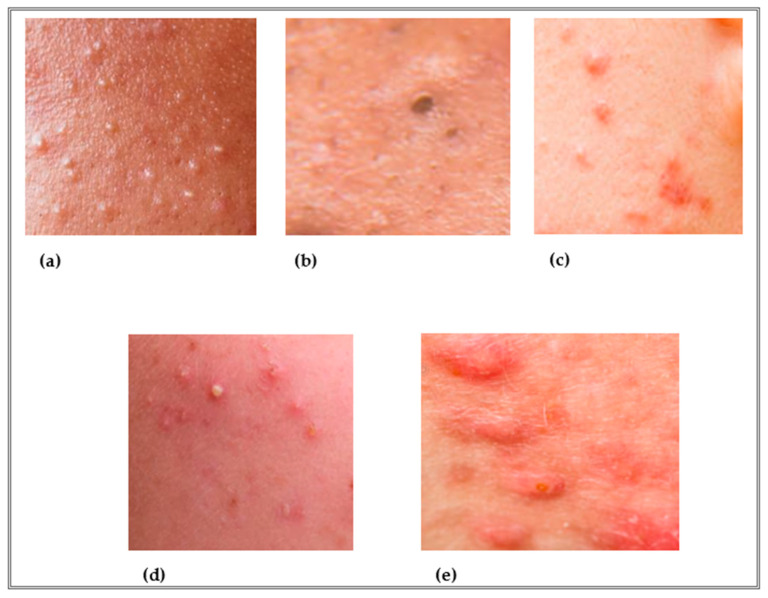
Acne vulgaris lesions. (**a**) Closed comedones; (**b**) Open comedones; (**c**) Skin papula; (**d**) Skin pustula; (**e**) Nodular acne. Adapted from: https://www.freepik.com/free-photos-vectors/acne (accessed on 20 April 2023).

**Figure 3 antioxidants-12-01264-f003:**
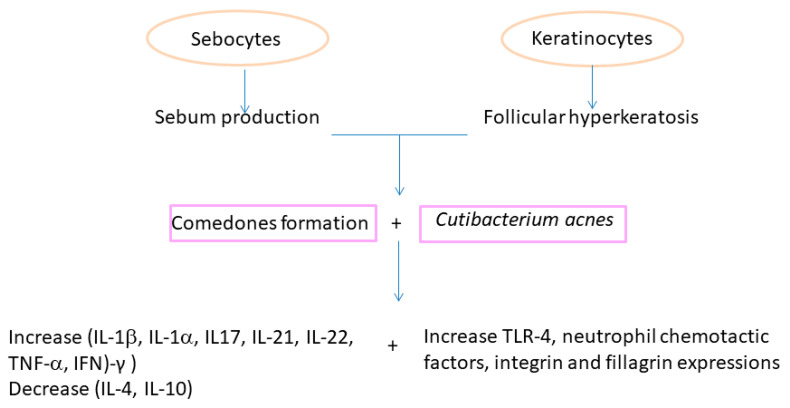
Schema of molecular mechanisms involved in the acne vulgaris.

**Figure 4 antioxidants-12-01264-f004:**
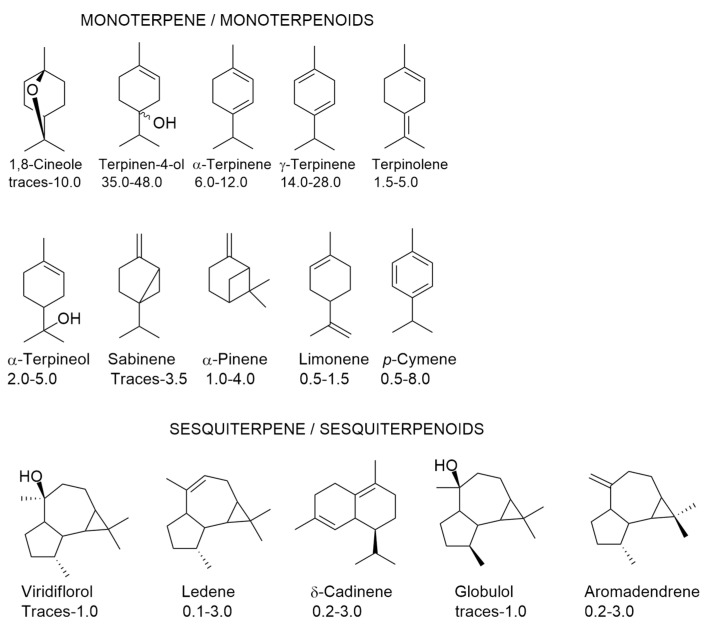
Chemical structures of the monoterpene/monoterpenoids and sesquiterpene/sesquiterpenoids present in the TTO and their concentrations in percentages [37].

**Table 1 antioxidants-12-01264-t001:** Antimicrobial activity, biofilm reduction, or biofilm inhibition induced by TTO, according to several works made by diverse research teams.

Main Components, Percentage (>5%)	Microorganism	Diameters of Inhibition Zones (mm)	Minimal Bactericidal Concentration (MBC) TTO	Minimal Inhibitory Concentration (MIC) TTO	Biofilm Reduction (R) or Inhibition (I) (Concentration)	Reference
Terpinen-4-ol, 42 Terpinen-4-ol standard	*Legionella pneumophila*	-	0.25–0.5% (*v/v*)	0.06–0.125% (*v/v*)	-	[59]
β-Pinene, 9 β-Terpineol, 6 Terpinen-4-ol, 10 α-Terpineol, 20	Planktonic *Enterococcus faecalis* *E. faecalis* biofilm inhibition	-	0.5%	0.25%	≥0.25% (I)	[60]
α-Terpinene, 9 γ-Terpinene, 20 Terpinen-4-ol, 43	*Escherichia coli* 22BT *E. coli* 45DT *Enterococcus faecium A29* * E. faecalis* VAN3 *Staphylococcus aureus* C3 *S. aureus* O	-	-	128 μg/mL 1 μg/mL 1 μg/mL 64 μg/mL 8 μg/mL 8 μg/mL	128 μg/mL (R) 1 μg/mL (R) 1 μg/mL (R) 64 μg/mL (R) 8 μg/mL (R) 8 μg/mL (R)	[61]
α-Terpinene, 11 γ-Terpinene, 19 Terpinen-4-ol, 33	14 Clinical and 2 references *S. aureus* strains	Liquid 8–30 Volatile 0–15	-	Liquid 0.1–0.8% (*v/v*) Liquid (biofilm)	Liquid (biofilm) 0.8–6.3% (*v/v*) (minimal biofilm eradication)	[62]
Area by standard GC-MS α-Pinene, 12 1,8-Cineole, 15 γ-Terpinene, 10 *o*-Cymene, 6 Terpinen-4-ol, 35 Area by Head Space GC-MS α-Pinene, 23 1,8-Cineole, 17 γ-Terpinene, 11 *o*-Cymene, 9 Terpinen-4-ol, 29	Methicillin-susceptibility *Staphylococcus aureus* Methicillin-resistant *Staphylococcus aureus* *Escherichia coli* Extended Spectrum Beta-Lactamases Carbapenem-Susceptible Kp Extended Spectrum Beta-Lactamases Carbapenem-Resistant *Acinetobacter baumannii* *Pseudomonas aeruginosa* Methicillin-susceptibility *Staphylococcus aureus +* oxacillin Methicillin-resistant *Staphylococcus aureus* + oxacillin			2% (*v/v*) 2% (*v/v*) 0.25 0.50 0.25 0.25 1	1% (*v/v*) 0.50% (*v/v*) 0.25% (*v/v*) 0.50% (*v/v*) 0.25% (*v/v*) 0.25% (*v/v*) 1% (*v/v*) Fractional inhibitory concentration index 0.32 (synergism) 0.32 (synergism)	[63]
Terpinen-4-ol, 40 γ-Terpinene, 12 1,8-Cineole, 7 *p*-Cymene, 6	* * *Bacillus subtilis * *Enterococcus faecalis * *Micrococcus luteus * *Staphylococcus aureus * *Pseudomonas aeruginosa * *Yersinia enterocolitica * *Salmonella enterica * *Serratia marcescens * *Pseudomonas fluorescens* (biofilm) *Salmonella enterica* (biofilm) *Candida albicans * *C. glabrata * *C. krusei * *C. tropicalis*	9.33 10.67 7.67 7.33 6.00 6.00 7.33 6.67 6.00 6.00 10.67 7.67 6.33 8.33		MIC 90 (μL/mL) 18.36 18.45 18.68 14.26 12.32 15.46 16.36 16.24 28.59 25.43 26.76 29.85 26.32 27.46		[64]
Terpinen-4-ol, 36	*Staphylococcus aureus * Coliform bacilli *Proteus* spp. *Klebsiella* spp. *Escherichia coli * *Citrobacter* spp. *Enterobacter* spp. *E. coli* (NCTC 11560) Fecal streptococci Fecal streptococci β-Hemolytic streptococci GP.2 *Enterococcus faecalis* (ATC29212) β-Hemolytic streptococci *Streptococcus pyogenes * Coagulase-negative staphylococci MRSA *Staphylococcus aureus* (NCTC 6571) *Candida* spp. *P. aeruginosa * *P. aeruginosa* (NCTC10662)		1 2 4 2 2 2 4 4 4 >8 >8 >8 >8 1–4 4 4 2 1 1–5 >8	0.5 1–2 2 1 1 1 2 2 2 >8 >8 8 >8 0.5–2 2 2–4 2 2 0.5 2–6 8		[65]
Without a chemical profile, only with the following information: TTO complied with the ISO 4730 and European Pharmacopoeia standards	Twenty-seven clinical isolates of *S. aureus* and the reference strain *S. aureus* NCTC 8325-4		0.25–1% (*v/v*) 0.5% (*v/v*)	0.125–0.5% (*v/v*) 0.5% (*v/v*)	2% (*v/v*) 1% (*v/v*)	[66]
TTO (enterprise 1) TTO (enterprise 2) Terpinen-4-ol (racemic) L-Terpinen-4-ol TTO (enterprise 1) TTO (enterprise 2) Terpinen-4-ol (racemic) L-Terpinen-4-ol	Thirty MRSA isolates Twenty-eight CoNS isolates		1–8% (*v/v*) 1->8% (*v/v*) 0.125–1% (*v/v*) 0.125–1% (*v/v*) 0.5–2% (*v/v*) 0.5–2% (*v/v*) 0.25–0.5% (*v/v*) 0.25–0.5% (*v/v*)	0.125–1% (*v/v*) 0.125–1% (*v/v*) 0.0625–0.5 (*v/v*) 0.0625–0.5 (*v/v*) 0.25–0.5% (*v/v*) 0.125–0.5% (*v/v*) 0.0625–0.25% (*v/v*) 0.0625–0.25% (*v/v*)		[67]
Terpinen-4-ol, 40 α-Terpinene, 9 γ-Terpinene, 21	*Staphylococcus aureus * * * *Escherichia coli * * * * * *Candida albicans*	1 mg/mL (16.57) 0.5 mg/mL (15.54) 0.01 mg/mL (11.08) 1 mg/mL (16.75) 0.5 mg/mL (15.13) 0.01 mg/mL (9.87) 0.01 mg/mL (12.21)				[68]
Terpinen-4-ol, 44 γ-Terpinene, 22 α-Terpinene, 7 α-Terpineol, 6	* * *C. albicans * *Trichophyton mentagrophytes * *S. aureus * *S. epidermidis * *Streptococcus pyogenes * MRSA *Klebsiella pneumoniae * *P. aeruginosa * * * *C. albicans * *Trichophyton mentagrophytes * *S. aureus * *S. epidermidis * *Streptococcus pyogenes * MRSA *Klebsiella pneumoniae * *P. aeruginosa*	TTO 20.3 21.1 19.2 21.7 19.2 19.5 18.1 13.2 AgNO_3_ 17.7 19.2 18.2 19.2 22.4 17.6 24.2 15.3				[69]
Not determined	*Bacteroides * *Prevotella * *Fusobacterium * *Peptostreptococcus anaerobius * Other gram-positive anaerobic cocci			0.03–0.5% (*v/v*) 0.03–0.25% (*v/v*) 0.06–0.55% (*v/v*) 0.06–0.25% (*v/v*) 0.03–0.25% (*v/v*)		[70]
Three batches Terpinen-4-ol, 41–44 α-Terpinene, 10–11 γ-Terpinene, 21–23	*Cutibacterium acnes*			0.25% (*v/v*)		[71]
Terpinen-4-ol, 40 1,8-Cineole, 5	*Trichophyton rubrum * *T. mentagrophytes * *Microsporum canis * *Candida albicans * *Candida* sp. *Trichosporon cutaneum * *Malassezia furfur* isolated from patients with Dandruff Seborrheic dermatitis Pityriasis versicolor			0.11–0.22% (*m/v*) 0.11–0.44% (*m/v*) 0.11% (*m/v*) 0.44% (*m/v*) 0.22–0.44% (*m/v*) 0.22% (*m/v*) 0.05–0.44% (*m/v*) 0.11–0.22% (*m/v*) 0.05–0.22% (*m/v*)		[72]
α-Terpinene, 9 γ-Terpinene, 19 Terpinen-4-ol, 46	*Chromobacterium violaceum* CV026	At MIC 0.25 mg/mL: 14.3		2 mg/mL		[73]
α-Terpinene, 10 *p*-Cymene, 24 Terpinen-4-ol, 25 β-Fenchyl alcohol, 9 Oregano + TTO TTO + Cinamom TTO + Lavender TTO + Laurel Oregano + TTO TTO + Cinamom TTO + Lavender TTO + Laurel Oregano + TTO TTO + Cinamom TTO + Lavender TTO + Laurel	*Streptococcus pyogenes* ATCC 19625 *Staphylococcus aureus* ATCC 25923 *Streptococcus agalactiae* ATCC 12386 * * *Streptococcus pyogenes* ATCC 19625 *S. aureus* ATCC 25923 *Streptococcus agalactiae* ATCC 12386	15.00 29.50 26.50 No interaction Additive effect Synergic effect No interaction Synergic effect No interaction No interaction Additive effect Synergic effect No interaction Synergic effect No interaction	2.00 0.25 Growth	1.00 0.125 1.00		[74]
*Cupressus sempervirens* + TTO *Myrtus communis* + TTO *Origanum marjorana* + TTO *Origanum vulgare* + TTO	*Streptococcus agalactiae* ATCC 55618 *S. pneumoniae* ATCC 49619 *S. pyogenes* ATCC 12344 *Mycobacterium smegmatis* ATCC 19420 *Moraxella catarrhalis* ATCC 23246 *Cryptococcus neoformans* ATCC 14116 *Staphylococcus aureus* ATCC 25924 *Streptococcus agalactiae* ATCC 55618 *S. pneumoniae* ATCC 49619 *S. pyogenes* ATCC 12344 *Mycobacterium smegmatis* ATCC 19420 *Klebsiella pneumoniae* ATCC 13883 *Moraxella catarrhalis* ATCC 23246 *Cryptococcus neoformans* ATCC 14116 *Staphylococcus aureus* ATCC 25924 *Streptococcus agalactiae* ATCC 55618 *S. pneumoniae* ATCC 49619 *Mycobacterium smegmatis* ATCC 19420 *Klebsiella pneumoniae* ATCC 13883 *Moraxella catarrhalis* ATCC 23246 *Cryptococcus neoformans* ATCC 14116 *Streptococcus agalactiae* ATCC 55618 *S. pneumoniae* ATCC 49619 *Mycobacterium smegmatis* ATCC 19420 *Klebsiella pneumoniae* ATCC 13883 *Cryptococcus neoformans* ATCC 14116	Additive effect Additive effect Additive effect Synergic effect Additive effect Synergic effect Synergic effect Additive effect Synergic effect Synergic effect Synergic effect Synergic effect Synergic effect Synergic effect Additive effect Additive effect Additive effect Synergic effect Synergic effect Synergic effect Additive effect Synergic effect Synergic effect Synergic effect Additive effect		1.00 mg/mL 2.00 mg/mL 1.50 mg/mL 1.00 mg/mL 2.00 mg/mL 0.09 mg/mL 2.00 mg/mL 1.00 mg/mL 1.00 mg/mL 2.00 mg/mL 1.00 mg/mL 1.00 mg/mL 0.25 mg/mL 2.00 mg/mL 1.00 mg/mL 1.50 mg/mL 3.00 mg/mL 1.00 mg/mL 1.00 mg/mL 0.25 mg/mL 1.00 mg/mL 1.00 mg/mL 2.00 mg/mL 1.00 mg/mL 0.50 mg/mL		[75]
γ-Terpinene, 17 4-Terpinenyl acetate, 67	*Cutibacterium acnes * *Staphylococcus epidermidis*		0.053 g/mL 0.053 g/mL	0.053 g/mL 0.053 g/mL	R: No effect I: 0.107 g/mL	[76]
The quantification of the components were not provided	*Staphylococcus aureus* strain EG-AE1 *Staphylococcus epidermidis* strain EG-AE2 *Cutibacterium acnes* Strain EG-AE1	15.5 21.02 20.85	78 mg/mL 78 mg/mL 39 mg/mL	78 mg/mL 78 mg/mL 39 mg/mL		[77]

**Table 2 antioxidants-12-01264-t002:** Characteristics and results of studies assessing the efficacy of tea tree oil on acne vulgaris in humans.

Author, Year of Publication	Study	Participants	Acne Severity	Product Application	Duration	Safety	Outcomes (End of Study)
Basset et al., 1990 [153]	RCT single-blind	*Intervention group* (n = 58): tea tree oil (TTO) 5% water-based gel. *Control group* (n = 61): Benzoyl Peroxide (BP) 5% water-based lotion. Mean age 19.7 years (range 12–35 years)	Mild-moderate acne vulgaris	Unmentioned	3 months	*The control group* reported more adverse effects (79%) than the *intervention group* (44%) (*p* < 0.001). Adverse effects reported: skin dryness, pruritus, stinging, burning, and redness. Dryness was the most reported.	Inflammatory lesions (IL): BP was significantly better than TTO in the reduction of the number of IL (*p* < 0.001). However, both treatments were effective in reducing IL. Non-inflammatory lesions (NIL): no significant differences between groups. Both showed a reduction of NIL (TTO group *p* < 0.05; BP group *p* < 0.01). Skin oiliness: BP showed increasingly less skin oiliness than the TTO group (*p* < 0.02).
Enshaieh et al., 2007 [154]	RCT double-blind	*Intervention group* (n = 30): TTO 5% carbomer gel. Mean age = 19.3 ± 3.1 years *Control group* (n = 30): carbomer gel only Mean age = 19.13 ± 2.64 years	Mild-moderate acne vulgaris	Applied twice daily over the affected area, for 20 min, wash with tap water	45 days	Minimal pruritus: 10% TTO group; 6.66% control group. Burning sensation on application: 3.33% TTO group; 6.66% control group. Minimal scaling: 3.33% TTO group.	Total Lesion Count (TLC): *Intervention group:* 43.64% reduction (significant, *p* = 0.035). *Control group:* 12.03% reduction (non-significant, *p* = 0.09). A significant difference between groups (*p* = 0.000). TTO 5% was 3.55 times more effective. Acne Severity Index (ASI): *Intervention group:* 40.49% reduction (significant, *p* = 0.000). *Control group:* 7.04% reduction (non-significant, *p* = 0.051). A significant difference between groups (*p* = 0.000). TTO 5% was 5.75 times more effective. Comedones: *Intervention group:* 40.24% reduction (significant, *p* = 0.000). *Control group:* 12.13% reduction (significant, *p* = 0.001). A significant difference between groups (*p* = 0.000). Papules: *Intervention group:* 46.06% reduction (significant, *p* = 0.004). *Control group:* 9.70% reduction (non-significant, *p* = 0.056). A significant difference between groups (*p* = 0.022). Pustules: *Intervention group:* 47.45% reduction (significant, *p* = 0.001). *Control group:* 2.37% increase (non-significant, *p* = 0.45). A significant difference between groups (*p* = 0.001).
Yadav et al., 2011 [160]	RCT uncontrolled, open-label multicentric phase III	*Group I* (n = 48): Oral tablets (Neem extract 200 mg + Turmeric extract 200 mg + Piper extract 10 mg). *Group II* (n = 46): Dermatological gel (TTO 5%). *Group III* (n = 47): Oral tablets + Dermatological gel. Age range: 15–50 years. 139 patients under 30 years	Mild-moderate acne vulgaris	Once daily on the affected area. (Group I and III: one tablet twice a day)	4 weeks	No serious adverse effects were reported.	*Group II (Gel TTO 5%) * Blackheads: 78.3% improvement (*p* < 0.05). Similar in all groups. Papules: 71.4% improvement (*p* < 0.05). Less improvement compared to other groups. Pustules: 86.4% improvement (*p* < 0.05). The greater improvement compared to other groups. Cysts: 8.8% improvement (*p* > 0.05). The greater improvement compared to other groups. Nodules: 13.0% improvement (*p* > 0.05). Less improvement compared to other groups. Group II showed 17% more improvement in deep inflammatory lesions compared to Group I and over 20% compared to Group III. The improvement observed with gel alone (group II) was slightly lower at the end of the study compared to the other groups.
Kim and Shin, 2013 [161]	Experimental pretest-posttest	*Intervention group* (n = 27): Mixture of TTO 3%, Lavender oil 2% and Jojoba oil + weekly acne treatment Mean age = 21.5 ± 2.2 years *Control group* (n = 27): weekly acne treatment Mean age = 20.9 ± 1.8 years	Not stated	Twice daily. Subjects were to keep the formulation on their skin for 5 min.	4 weeks	One subject (n = 1) from the *intervention group* complained of itch 2–3 days after beginning the treatment, which resolved shortly without subsequent adverse effects.	The number of *C. acnes* on the foreheads (t = 3.100, *p* =0.005) and the total number of *C. acnes* (t = 3.061, *p* = 0.005) significantly reduced at the end of the intervention. IL: significantly reduced only in the intervention group (t = 5.544, *p* < 0.001). NIL: significantly reduced in both, intervention (t = 3.406, *p* = 0.002) and control (t = 3.257, *p* = 0.003) groups. TLC: significantly reduced in both groups (intervention: t = 6.537, *p* < 0.001; control: t = 2.947, *p* =0.007). Sebum excretion: significant reduction in the sebum excretion rate (t = 3.144, *p* = 0.004) only in the intervention group.
Kwon et al., 2014 [155]	RCT double-blind split-face	n = 32 One side: 5% Lactobacillus-fermented *Chamaecyparis obtusa* (LFCO) Other side: 5% TTO extract to the other side. Mean age = 25.9 ± 5.6 years	Mild-moderate acne vulgaris	Twice daily.	8 weeks	No severe adverse reactions. *TTO side:* 12.5% mild dryness and 18.8% mild erythema and desquamation. *LFCO side:* 6.3% transient mild erythema and 6.3% dryness	IL: significantly reduced on the LFCO side (65.3%, *p* < 0,01) and TTO side (38.2%, *p* < 0.01). A significant difference in the mean of IL counts between LFCO and TTO sides (*p* < 0.05) NIL: significantly reduced on the LFCO side (52.6%, *p* < 0.01) and TTO side (23.7%, *p* < 0.05). A significant difference in the mean of NIL counts between LFCO and TTO sides (*p* < 0.05) Acne Grade (Leeds revised acne grading): significantly decreased on both sides (LFCO side from 4.0 to 1.8, *p* < 0.01; TTO side from 4.0 to 2.9. *p* < 0.01). Sebum secretion: LFCO side showed a significant decreased (*p* < 0.05) but the TTO side did not. Sebaceous gland size: significantly decreased size of the sebaceous gland on the LFCO side (*p* = 0.03), but not on the TTO side. Proteins related to inflammation: significant decrease on both sides. SREBP-1 and IGF-1R expression showed reductions only on the LFCO side (*p* < 0.05). IL-8 and TLR-2 mRNA expression were significantly reduced on both sides, but greater on the LFCO side.
Malhi et al., 2016 [156]	Dual-centre, open-label, phase II pilot study	n = 14 Tea Tree Medicated Gel (200 mg/g) and Tea Tree Face Wash for Acne (7 mg/g). Mean age = 26 ± 7 years	Mild-moderate acne vulgaris	Twice daily. Wash with one pump of the face wash, then apply a pea-sized gel. Leave the product on for at least 6 h and wash it off only at the next application time.	12 weeks	No serious adverse reaction. Moderate scaling, peeling, and dryness were recorded at week 4 and reduced at week 12.	TLC: significant decrease of 54% (*p* < 0.001). IGA Score (Investigator Global Assessment): significant decrease from a mean of 2.4 to 1.9 (*p* < 0.05). Facial oiliness: mean score significantly decreased from 2.0 to 1.1 (*p* < 0.01). Clinical efficacy: defined as a reduction in TLC ≥ 40% at the end of the study. Products were clinically effective in 79% of the participants.
Mazzarello et al., 2018 [159]	Single-center, randomized, double-blinded, comparative study	*Intervention group* 1 (n = 20): 20% propolis extract, 3% tea tree oil, and 10% aloe vera leaf juice cream (PTAC). Mean age = 27 ± 7.44 years *Intervention group* 2 (n = 20): 3% erythromycin cream (EC) Mean age = 23 ± 5.06 years *Control group* (n = 20): placebo Mean age = 24 ± 6.14 years	Mild-moderate acne vulgaris	Twice daily.	30 days	Not stated	Sebum, pH, and erythema index values (healthy skin) did not show statistically significant changes in the three groups. The Erythema index of scars and papules showed a statistically significant difference in PTAC and EC groups (*p* < 0.001). Comedones: Significant reduction (*p* < 0.001) in PTAC (38.2%) and EC (42.1%) groups. Papules: Significant reduction (*p* < 0.001) in PTAC (61.4%) and EC (45.4%) groups, and in placebo (4.6%, *p* < 0.05) Pustules: Significant reduction (*p* < 0.001) in PTAC (58.7%) and EC (44.1%) groups. ASI: Significant reduction (*p* < 0.001) in PTAC (66.7%) and EC (49.7%) groups. TLC: Significant reduction (*p* < 0.001) in PTAC (63.7%) and EC (46.5%) groups, and in placebo (9.9%, *p* < 0.05)
Lubtikulthum et al., 2019 [157]	Observer-blinded, noninferiority randomized controlled study	*Intervention group* (n = 38): Topical Herbal Extract Formula (HEF). A gel that contains onion, Lavandula, mangosteen, aloe vera, paper mulberry, and tea tree extracts. Mean age = 21.79 ± 2.238 years *Control group* (n = 36): BP 2.5% cream Mean age = 21.89 ± 2.153 years	Mild-moderate acne vulgaris	Twice daily. Washed off 15 min after application.	12 weeks	No systemic side effects occurred. The most common adverse effect was skin irritation. Transient itching of acne lesions and burning sensations were reported after the application of both products. Week 2: glazing with peeling and cracking was less common in the *intervention group* (12.82%) than in the *control group* (28.95%). Erythema with minimal edema or minimal papular response only was reported in the *control group* (7.89%).	Comedones: *Intervention group:* 34.51 ± 31,01%, a significant reduction. *Control group:* 39.4 ± 2.18%, a significant reduction. IL: *Intervention group:* 40.54 ± 44.75%, a significant reduction. *Control group:* 45.3 ± 35.68%, a significant reduction. TLC: *Intervention group:* 36.47 ± 30.1%, a significant reduction. *Control group:* 40.9 ± 21.67%, a significant reduction. The mean difference between groups in the percent reduction in comedones, IL and TLC at the end of the study showed an inconclusive result for noninferiority
Najafi-Taher, 2022 [158]	RCT, triple-blind	*Intervention group* (n = 53): tea tree oil (6%) nanoemulsion containing adapalene gel 0.1% (TTO + ADA) Mean age = 26.72 ± 5.231 years *Control group* (n = 47): adapalene gel 0.1% (ADA) Mean age = 27.36 ± 5.036 years	Mild-moderate acne vulgaris	Once a day (at night). Applied to clean and dry skin on affected areas.	12 weeks	Mild severity topical adverse reaction was reported in both groups. Dryness was de most frequently reported in both groups. Other reactions reported war irritation, erythema, and burning sensation.	IL and NIL: Decreased in both groups, but a greater decrease in TTO + ADA. Statistically difference between groups (*p* < 0.001) ASI: Significant reduction (*p* < 0.001) in both groups, but more obvious for TTO + ADA with 71.69% treatment success compared with 6.38% in the ADA group. TLC: Decreased in both with significant difference between groups (*p* < 0.001). TTO + ADA showed a greater decrease.
Lupu et al., 2022 [162]	Experimental blinded study	*Intervention group* (n = 35) Application of two products: Ivapur purifying cleansing gel (among other ingredients, zincidone^®^, glycerine, and Herculane spring thermal water) and Ivapur K cream (among other ingredients, lactic acid, salicylic acid, piroctone olamine, *tea tree oil*, bisabolol, and Herculane spring thermal water) Mean age = 19.17 ± 6.37 years	Oily, acne-prone skin	Twice daily. Clean with purifying cleansing gel and then applied the cream.	28 days	No adverse effects were reported.	Facial oiliness: mean score significantly decreased by 2.05 points (*p* < 0.001). *C. acnes* colonization: the number of follicles colonized by *C. acnes* was significantly reduced by an average of 72 (*p* = 0.003). At the end of the study was observed decrease in the number of infundibula with thickened bright borders (*p* < 0.001), the follicles with dilated infundibula (*p* = 0.005), the comedones (*p* < 0.001), and in the inflammatory infiltrate density (*p* < 0.001). On the opposite, the number of regular follicles increased (*p* < 0.001)

ADA: adapalene; ASI: Acne Severity Index; BP: Benzoyl Peroxide; EC: Erythromycin cream; HEF: Herbal Extract Formula; IGF-1R: Insulin-like Growth Factor 1 Receptor; IL: Inflammatory Lesions; LFCO: Lactobacillus-fermented *Chamaecyparis obtuse*; NIL: Non-Inflammatory Lesions; PTAC: Propolis extract, Tea tree oil, and Aloe vera leaf juice Cream; SREBP-1: Sterol Regulatory Element-Binding Protein 1; TLC: Total Lesions Count; TLR-2: Toll-Like Receptor-2; TTO: Tea Tree Oil.

## Data Availability

Not applicable.

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
