# Peer review of "Tea Tree Oil: Properties and the Therapeutic Approach to Acne—A Review"

_antioxidants, 2023, doi:10.3390/antiox12061264_

Round 1

Reviewer 1 Report

line 74: revise "have an antibacterial effect has against"

line 84: I cannot understand "or expression in citrus fruit"

line 520: change to "in in vitro"

Text regarding in vitro studies are too generic. The authors must indicate the dose/concentration for each biological activity or enzyme inhibition, otherwise it is just  alist of activities which are not of interest to the readers who are compelled to download every article to know more about the potency.

it is ok

Author Response

Dear Assistant Editor,

Ms. Nova Tang,

Date: 01/06/2023

Subject: Antoxidants Decision Letter Reply

Dear Ms. Nova Tang,

I thank you for your e-mail of 22 May 2022 with the Decision Letter on Manuscript ID: antioxidants-2407472, titled: Tea tree oil: properties and the therapeutic approach to acne. A review.

I have read it carefully and I do understand most of the referee’s comments. Please find below our reply and comments addressing each point raised by the reviewer. All corrections, additions and changes performed in the MS and they are marked with yellow colour.

Reviewer 1

Comments and Suggestions for Authors

  • line 74: revise "have an antibacterial effect has against"

It was corrected. Thank you for the suggestion

  • line 84: I cannot understand "or expression in citrus fruit"

It was modified in order to better understand the procedure used.

  • line 520: change to "in in vitro"

It was corrected.

  • Text regarding in vitro studies are too generic. The authors must indicate the dose/concentration for each biological activity or enzyme inhibition, otherwise it is just  alist of activities which are not of interest to the readers who are compelled to download every article to know more about the potency.

Thank you for the suggestion. It was taken into account for almost cases.

  • Comments on the Quality of English Language

Reviewer 2

  • The topic covered by the authors of the manuscript seems interesting and has a practical aspect. The scientific material is presented in a solid way and does not require significant revision. Below are some minor comments that I suggest you take into consideration:
  • I propose that the abbreviation alternifolia be used in the text of the manuscript instead of Melaleuca alternifolia;

Thank you for the suggestion. It was corrected.

  • full binominal botanical names should be given for all plant species cited, e.g. Lavandula angustifolia or Curcuma longa L.;

Thank you for the suggestion. It was added.

  • in line 85, instead of "anticancer" I suggest using the more suitable term antineoplastic (few plant components are practically used in cancer therapy);

Thank you for the suggestion. It was corrected.

  • if possible, please supplement the description of TTO activity with information on the concentration of the oil that was evaluated in the given range of biological activity, as well as information on the chemical composition (% content of the main monoterpenes) of these preparations; perhaps it was a pure oil and not its solution?

Thank you for the suggestion. It was added in almost cases.

  • no information on MIC/MBC for many pathogens;

Thank you for the suggestion. It was added in almost cases.

  • line 519 (e.g. vitamin E, vitamin C, phenols, essential oils), compared to essential oils, the more well-known natural antioxidants are carotenoids, most essential oils show antimicrobial properties and the antioxidant effect per se among monoterpenes is rarer;

Thank you for the analysis, but these metabolites are introduced because they are reported in the references cited. For this reason, I left them in the text. Carotenoids are important singlet oxygen scavengers. Vitamins E and C work together in the organisms. Vitamin E, after its action as an antioxidant remains as tocopheryl radical (oxidized form) and it is regenerated to vitamin E after the action of vitamin C. Carnosic acid is a phenolic diterpene found in rosemary and it is a strong antioxidant used in numerous industrial and medicinal/pharmaceutical applications. Regarding essential oils, those presenting some volatiles (thymol, carvacrol, p-cymene…) show antioxidant activity. Thymol and carvacrol are phenolic monoterpenes that can be found in diverse species of the genus Thymus and Origanum Thus, and with due respect to the reviewer, we would prefer to leave those compounds in the text.

  • line 530 (At the end of the extraction, the surfactant nonionic Triton 530 CG-110 does not remain in the TTO because it is volatile), the sentence is confusing, after all, TTO is also a volatile substance;

Thank you for calling our attention to this confusing sentence. We have modified. At the end of the extraction, the surfactant nonionic Triton CG-110 does not remain in the TTO because it is much less volatile than TTO

  • line 573 (TOO), probably a misspelling;

Thank you for the suggestion. It was corrected.

  • line 577 (Dihydroethidium), the names of compounds in the middle of a sentence should be written in lowercase;

Thank you for the suggestion. It was corrected.

  • other minor editorial errors should be corrected.

We have corrected other small errors that we found in the text after re-writing

Reviewer 3

  • The manuscript is a review on the properties of Tea tree oil.
  • The authors should justify the goal of the review on Tea tree oil and described the tools used to choose the discussed articles.

We added the objective of the literature review at the end of the introduction (line 111-115). Since this is a narrative review, the selected articles were all those that met the proposed objectives, ensuring the relevance of the information. In the case of the articles included in the human trials, the selection methodology was described at the beginning of chapter 7 (line 1126-1132)

  • What is the purpose of this review?

Thank you for the question. The aim of the review, as mentioned above, was defined at the end of the introduction (line 111-115). In summary, it was our aim to describe the various properties of TTO that are vastly studied in vitro and to subsequently present its efficacy in the treatment of acne with evidence from human trial results, because as we know, positive laboratory results are not always reflected in beneficial clinical outcomes. We hope to have responding to the question.

  • To whom is this review addressed? It is not clear which scientific community the authors are addressing (physicians, pharmacologists, or others....)

Thank you for the question. Essentially health professionals who deal clinically with acne, so that they are aware not only of the results of human trials, but also of the characterization of the TTO and its properties in general, because, as stated in the review, TTO has various applications in areas other than acne. Thus, the description of the properties analysed in the laboratory seems important to us, in order to provide professionals with knowledge and to enable new clinical research with TTO. We hope to have responding to the question.

  • This review needs to be restructured. For example, sub-chapters could be inserted in order to better focus on the topic and purpose.

Thank you for the suggestion. We have included several sub-chapters throughout the review to make it easier to read.

  • It might be useful to separate the description about topic or oral administration use of TTO.
  • Thank you for the suggestion. We have separated by topic.

  • There are no toxicology data/adverse effects of Melaleuca alternifolia tea tree oil, why are they not reported?

Thank you for the suggestion. We have included a new chapter (chapter 6, line 1090-1123) on the toxicity of TTO

  • Line 86- After the introduction, make a separate, specific chapter on this plant Melaleuca alternifolia and report on the structures of the compounds listed below (lines 99-104) and the role of TTOs.

It was introduced.

  • Line 330 - " 4. Anti-inflammatory properties' - make a more explanatory title.

Thank you for the suggestion. We have renamed the chapter: " Anti-inflammatory properties of TTO".

  • Line 491  - “5. Antioxidant properties”- make a more explanatory title.

Thank you for the suggestion. We have renamed the chapter: "Antioxidant properties of TTO".

  • Authors often use the generic term "TTO" and do not specify which compound(s) the work refers to. This is especially important when discussing the different results reported in different papers, likely caused by qualitative and quantitative compositional variation in TTO.

This was addressed, particularly in the antimicrobial activity, in which it is possible to see the main volatiles present in TTO, nevertheless, it is scarce to find the correlation between activities and compounds. Generally and in what concerns phytotherapy, the extracts or in this case EOs, the activities found for these extracts are better than a sole compound. The explanation provided by many researchers is the synergism effect between the compounds present in the extracts or EOs.

I hope that I have adequately addressed the reviewer remarks and questions, and that the manuscript is now suitable for publication.

Yours sincerely,

Maria da Graça Miguel

Reviewer 2 Report

The topic covered by the authors of the manuscript seems interesting and has a practical aspect. The scientific material is presented in a solid way and does not require significant revision. Below are some minor comments that I suggest you take into consideration:

- I propose that the abbreviation M. alternifolia be used in the text of the manuscript instead of Melaleuca alternifolia;

- full binominal botanical names should be given for all plant species cited, e.g. Lavandula angustifolia Mill. or Curcuma longa L.;

- in line 85, instead of "anticancer" I suggest using the more suitable term antineoplastic (few plant components are practically used in cancer therapy);

- if possible, please supplement the description of TTO activity with information on the concentration of the oil that was evaluated in the given range of biological activity, as well as information on the chemical composition (% content of the main monoterpenes) of these preparations; perhaps it was a pure oil and not its solution?

- no information on MIC/MBC for many pathogens;

- line 519 (e.g. vitamin E, vitamin C, phenols, essential oils), compared to essential oils, the more well-known natural antioxidants are carotenoids, most essential oils show antimicrobial properties and the antioxidant effect per se among monoterpenes is rarer;

- line 530 (At the end of the extraction, the surfactant nonionic Triton 530 CG-110 does not remain in the TTO because it is volatile), the sentence is confusing, after all, TTO is also a volatile substance;

- line 573 (TOO), probably a misspelling;

- line 577 (Dihydroethidium), the names of compounds in the middle of a sentence should be written in lowercase;

- other minor editorial errors should be corrected.

Author Response

(The authors gave the same response as above.)

Reviewer 3 Report

 The manuscript is a review on the properties of Tea tree oil.

The authors should justify the goal of the review on Tea tree oil and described the tools used to choose the discussed articles.

What is the purpose of this review?

To whom is this review addressed? It is not clear which scientific community the authors are addressing (physicians, pharmacologists, or others....)

This review needs to be restructured. For example, sub-chapters could be inserted in order to better focus on the topic and purpose.

It might be useful to separate the description about topic or oral administration use of TTO.

There are no toxicology data/adverse effects of Melaleuca alternifolia tea tree oil, why are they not reported?

Line 86- After the introduction, make a separate, specific chapter on this plant Melaleuca alternifolia and report on the structures of the compounds listed below (lines 99-104) and the role of TTOs.

Line 330 - " 4. Anti-inflammatory properties' - make a more explanatory title.

 Line 491  - “5. Antioxidant properties”- make a more explanatory title.

Authors often use the generic term "TTO" and do not specify which compound(s) the work refers to. This is especially important when discussing the different results reported in different papers, likely caused by qualitative and quantitative compositional variation in TTO.

Author Response

(The authors gave the same response as above.)

Round 2

Reviewer 1 Report

line 276: Klebsiella

line 1113: TTT?

Author Response

Dear Assistant Editor,

Ms. Sandy Li,

Date: 05/06/2023

Subject: Antoxidants Decision Letter Reply

Dear Ms. Sandy Li,

I thank you for your e-mail of 04 May 2023 with the Decision Letter on Manuscript ID: antioxidants-2407472, titled: Tea tree oil: properties and the therapeutic approach to acne. A review.

I have read it carefully and I do understand most of the referee’s comments. All corrections in the MS are marked with blue colour.

Reviewer 1

Comments and Suggestions for Authors

line 276: Klebsiella. It was corrected. Thank you.

line 1113: TTT? It was corrected for TTO. Thank you.

Thank you for the final decision.

Reviewer 3

Thank you for the decision of the reviewer.

I hope that I have adequately addressed the reviewer remarks and questions, and that the manuscript is now suitable for publication.

Yours sincerely,

Maria da Graça Miguel

Reviewer 3 Report

The authors considered the comments and the new revised version of the manuscript was improved.

The manuscript can be accepted in this revised form.

Author Response

(The authors gave the same response as above.)
